



# Structural changes to forests during regeneration affect water flux partitioning, water ages and hydrological connectivity: Insights from tracer-aided ecohydrological modelling

Aaron J. Neill[1], Christian Birkel[2,1], Marco P. Maneta[3,4], Doerthe Tetzlaff[5,6,1], Chris Soulsby[1,5]

5   [1] Northern Rivers Institute, University of Aberdeen, Aberdeen, United Kingdom.
   [2] Department of Geography, University of Costa Rica, San Pedro, Costa Rica.
   [3] Geosciences Department, University of Montana, Missoula, MT, USA.
   [4] Department of Ecosystem and Conservation Sciences, W.A Franke College of Forestry and Conservation, University of Montana, Missoula, MT, USA.
10  [5] Department of Ecohydrology, IGB Leibniz Institute of Freshwater Ecology and Inland Fisheries, Berlin, Germany.
   [6] Department of Geography, Humboldt University of Berlin, Berlin, Germany.

*Correspondence to*: Aaron J. Neill (aaron.neill@abdn.ac.uk)

**Abstract.** Increasing rates of biodiversity loss are adding momentum to efforts seeking to restore or rewild degraded landscapes. Here, we investigated the effects of natural forest regeneration on water flux partitioning, water ages and 15  hydrological connectivity, using the tracer-aided ecohydrological model EcH$_2$O-iso. The model was calibrated using ~3.5 years of diverse ecohydrological and isotope datasets available for a catchment in the Scottish Highlands, an area where the impetus for regeneration of native pinewoods is growing. We then simulated two land cover change scenarios that incorporated forests at early (thicket) and late (old-open forest) stages of regeneration, respectively, and compared these to a present-day baseline simulation. Changes to forest structure (proportional vegetation cover, vegetation heights and leaf area index of pine 20  trees) were modelled for each stage. Establishment of thicket forest had the greatest effect on water partitioning/ages and connectivity, with increased losses to interception evaporation driving reductions in below-canopy fluxes (soil evaporation, groundwater recharge and streamflow) and generally slower rates of water turnover. Effects on streamflow were most evident for low and moderate summer flows rather than winter high flows. Whilst full forest regeneration was limited to hillslopes, resultant changes to the spatial dynamics of flux partitioning could also cause drying out of the valley bottom. The more open 25  nature of the older forest generally resulted in water fluxes, ages and connectivity characteristics returning towards baseline conditions. Our work implies that the ecohydrological consequences of natural forest regeneration on degraded land depend on the structural characteristics of the forest at different stages of development. Consequently, future land cover change investigations need to move beyond consideration of simple forest vs. non-forest scenarios to inform management that effectively balances landscape restoration with demand for ecosystem services. Tracer-aided ecohydrological models were 30  also shown to be useful tools for land cover change simulations and further potential of such models was highlighted.





## 1 Introduction

Increasing rates of biodiversity loss and ecosystem degradation have highlighted the urgent need for landscape conservation and restoration (Rands et al., 2010). Unlike approaches seeking to retain sets of predetermined characteristics, rewilding takes a relatively "hands-off" approach to restoration by seeking to restore dynamic natural processes that create self-sustaining,

complex ecosystems (Navarro and Pereira, 2015; Perino et al., 2019). The Scottish Highlands represent a degraded landscape for which rewilding is increasingly promoted as a means of restoring native pinewoods that have been lost due to human land management practices (Deary and Warren, 2017; zu Ermgassen et al., 2018). Following the last glaciation, the dominant vegetation over much of the Highlands was open forests dominated by Scots Pine (*Pinus sylvestris*) and birch (*Betula* spp.) (Steven and Carlisle, 1959). However, industrial exploitation in the 17-19$^{th}$ centuries, and interruption of natural regeneration

due to intensification of sheep grazing and management of Highland estates for deer and grouse shooting since the mid-19$^{th}$ century (Steven and Carlisle, 1959; Wilson, 2015), means that remaining native pinewoods now cover only ~1% of their Holocene maximum extent (Mason et al., 2004). The goal of rewilding in the Highlands would be to restore the process of natural forest regeneration through initial human interventions to reduce grazing pressures and establish new seed sources through targeted tree planting, ultimately leading to the proliferation and maintenance of self-sustaining native pinewoods

(Thomas et al., 2015; zu Ermgassen et al., 2018).

Vegetation plays a crucial role in partitioning land-surface water and energy fluxes, whilst soil moisture determines water availability for root uptake and plant growth (Rodriguez-Iturbe, 2000), and determines the water-limited edge of forest extents (Simeone et al., 2018). Therefore, elucidating the potential ecohydrological consequences of natural forest regeneration is

crucial for sustainable land management and for understanding how land cover change may affect other ecosystem services. This is relevant beyond Scotland as reforestation is widely seen as a means of reducing flood and erosion risks, improving water quality, and mitigating climate change (Bonan, 2008; Chandler et al., 2018; Ellison et al., 2017; Iacob et al., 2017; Rudel et al., 2020). Of particular importance is how partitioning of water between "blue" (i.e. groundwater recharge and stream discharge) and "green" (i.e. evapotranspiration [ET]) fluxes is affected in space and time, as this has implications for water

availability to terrestrial and aquatic ecosystems, and downstream water users (Falkenmark and Rockström, 2006; 2010). In addition, consideration of water ages and the spatio-temporal dynamics of hydrological connectivity can reveal how storage-flux dynamics and hydrological source areas are affected by regeneration (Bergstrom et al., 2016; Kuppel et al., 2020; Tetzlaff et al., 2014; Sprenger et al., 2019). This has implications for ecosystem resilience to climatic extremes (Fennell et al., 2020; Kleine et al., 2020; Smith et al., 2020), generation of low/high flows (Birkel et al., 2015; Nippgen et al., 2015), and

redistribution of water and solutes (Bergstrom et al., 2016; Turnbull and Wainwright, 2019).

Previous work investigating the hydrological consequences of forest (re)generation has often employed the paired-catchment approach to assess how changes in forest cover affect aggregated metrics (e.g. water balance and water yield) that characterise





catchment functioning (Bosch and Hewlett, 1982; Brown et al., 2005; Filoso et al., 2017). However, findings from such studies

may be biased as many only consider the early stages (~first 10 years) of forest development, often within the context of commercial (possibly non-native) plantation management (Coble et al., 2020; Ellison et al., 2017; Filoso et al., 2017). Where long-term sites have been established, data have indicated that age-related changes to forest structure and tree physiology can substantially influence water partitioning (Coble et al., 2020; Marc and Robinson, 2007; Perry and Jones, 2017; Scott and Prinsloo, 2008; Segura et al., 2020). However, the focus on commercial plantations, especially in the UK context (Marc and

Robinson, 2007), may limit transferability of findings to scenarios of passively managed natural forest regeneration associated with rewilding. In particular, forest harvesting cycles (~40 years) are much shorter than the lifespan (>150 years) of trees in natural forests (Brown et al., 2005; Ellison et al., 2017; Summers et al., 2008), whilst plantation management practices (e.g. drainage, species selection, thinning, etc.) may confound effects of land cover change (Birkinshaw et al., 2014; Robinson, 1998). Along with general drawbacks to the paired-catchment approach (e.g. limited ability to resolve spatio-temporal changes

to internal catchment processes; Brown et al., 2005), these factors demonstrate the need to better understand the ecohydrological consequences of a naturally regenerating forest, using methods that can disaggregate the drivers of aggregated catchment responses in space and time.

Spatially distributed ecohydrological models explicitly simulate the tight coupling of water, energy, and vegetation dynamics

in time and space (Fatichi et al., 2016). Consequently, they are promising tools for investigating the ecohydrological impacts of land cover change (Ellison et al., 2017; Manoli et al., 2018; Peng et al., 2016). Models are also advantageous in providing a virtual, controlled environment within which different scenarios of land cover change can be simulated and compared against a baseline (Du et al., 2016). A critical prerequisite to using ecohydrological models is confidence in accurate representation of internal catchment functioning (Seibert and van Meerveld, 2016). Given that the integration of stable water isotope tracers

($\delta^2$H and $\delta^{18}$O) within models can have significant value in this regard (Birkel and Soulsby, 2015), the tracer-aided ecohydrological model EcH$_2$O-iso has recently been developed (Kuppel et al., 2018a). EcH$_2$O-iso has successfully been applied to a range of environments to elucidate links between land cover and water partitioning/ages (e.g. Douinot et al., 2018; Gillefalk et al., 2021; Knighton et al., 2020; Kuppel et al., 2020; Smith et al., 2019; 2020).

Here, we applied EcH$_2$O-iso to a small experimental catchment in the Scottish Highlands to investigate the ecohydrological consequences of natural pinewood regeneration on degraded land. Specifically, we compared a present-day baseline simulation with two land cover change scenarios that incorporated forests at early (thicket) and late (old-open forest) stages of regeneration, respectively. Changes to forest structure (proportional vegetation cover, vegetation heights and leaf area index of pine trees) were modelled for each stage. Soil properties were held constant as it is unclear how "effective" parameters

describing their aggregated characteristics respond to land cover change (Seibert and van Meerveld, 2016). Furthermore, conifer forests can have inconsistent effects on soil properties, as soil acidification from needle decomposition may compete with improvements to soil structure caused by increases in organic matter and root density (Archer et al., 2013; Chappell et





al., 1996). The wet and windy climate of Scotland also makes it likely that changes in canopy structure and interception losses will predominantly determine variations in water partitioning (c.f. Farley et al., 2005; Marc and Robinson, 2007). Our specific

objectives were to evaluate the effect of forest regeneration stage on:

1. Dynamics of water flux partitioning in time and space,
2. Ages of "blue" and "green" water fluxes,
3. Hydrological connectivity under contrasting flow conditions.

**2 Study site**

The Bruntland Burn (BB) catchment (3.2 km$^2$) is in the Cairngorms National Park in the Scottish Highlands (Fig. 1a). It is a tributary of the Girnock Burn catchment (31 km$^2$) that drains into the River Dee. The Dee supports a globally important Atlantic Salmon population and provides drinking water to over 300,000 people (Langan et al., 1997; Soulsby et al., 2016). The glacial legacy of the BB has left steep hillslopes and a flat valley bottom. Bedrock is mostly granite with schists and other metamorphic

rocks fringing the catchment. This is overlain by extensive drift deposits (70% of catchment area) that are 5-10 m deep on lower hillslopes and up to 40 m deep in the valley bottom (Soulsby et al., 2016). Peat (up to 4 m deep) and peaty gley soils overlay the deeper drift deposits with peaty podzols and poorly developed rankers characterising higher elevations along with some bedrock outcrops (Fig. 1a).

Natural Scots pine regeneration is restricted due to grazing from high densities of red deer (*Cervus elaphus*; 11 to 14.9 deer km$^{-2}$ [SNH, 2016]) and controlled burning of grouse moorlands. Consequently, tree cover is largely limited to native pinewoods on the relatively inaccessible steep northern hillslope and pine plantations at the catchment outlet (Fig. 1b-c). Vegetation otherwise reflects soil type; heather (*Calluna vulgaris, Erica tetralix*) dominates the peaty podzols and rankers of the hillslopes (Fig. 1d), whilst *Molinia* grassland on the peaty gleys (Fig. 1f) is increasingly outcompeted by *Sphagnum* spp. on the

waterlogged peats of the valley bottom (Fig. 1e). Isolated pine trees are scattered throughout the catchment, with those in the wetter valley bottom exhibiting stunted growth ("Bog pine" – Fig. 1g).

Mean annual precipitation and potential evapotranspiration are 1000 mm and 400 mm, respectively, with the former usually falling in low-intensity events (<10 mm d$^{-1}$). Less than 5% of precipitation falls as snow, reflecting mean temperatures ranging

between 1 ℃ and 13 ℃ in winter and summer, respectively. Seepage of fracture flow from bedrock outcrops and shallow sub-surface flow through the rankers predominantly move vertically on reaching the podzols to recharge stores of groundwater (GW) in the underlying drift (Blumstock et al., 2016; Tetzlaff et al., 2014) which sustain baseflow conditions in the stream (Blumstock et al., 2015). The storm response of the BB is non-linear, depending on the dynamic expansion of the riparian





saturation area which generates overland flow and hydrological connectivity between the hillslopes and valley bottom (Birkel

et al., 2015; Soulsby et al., 2015).

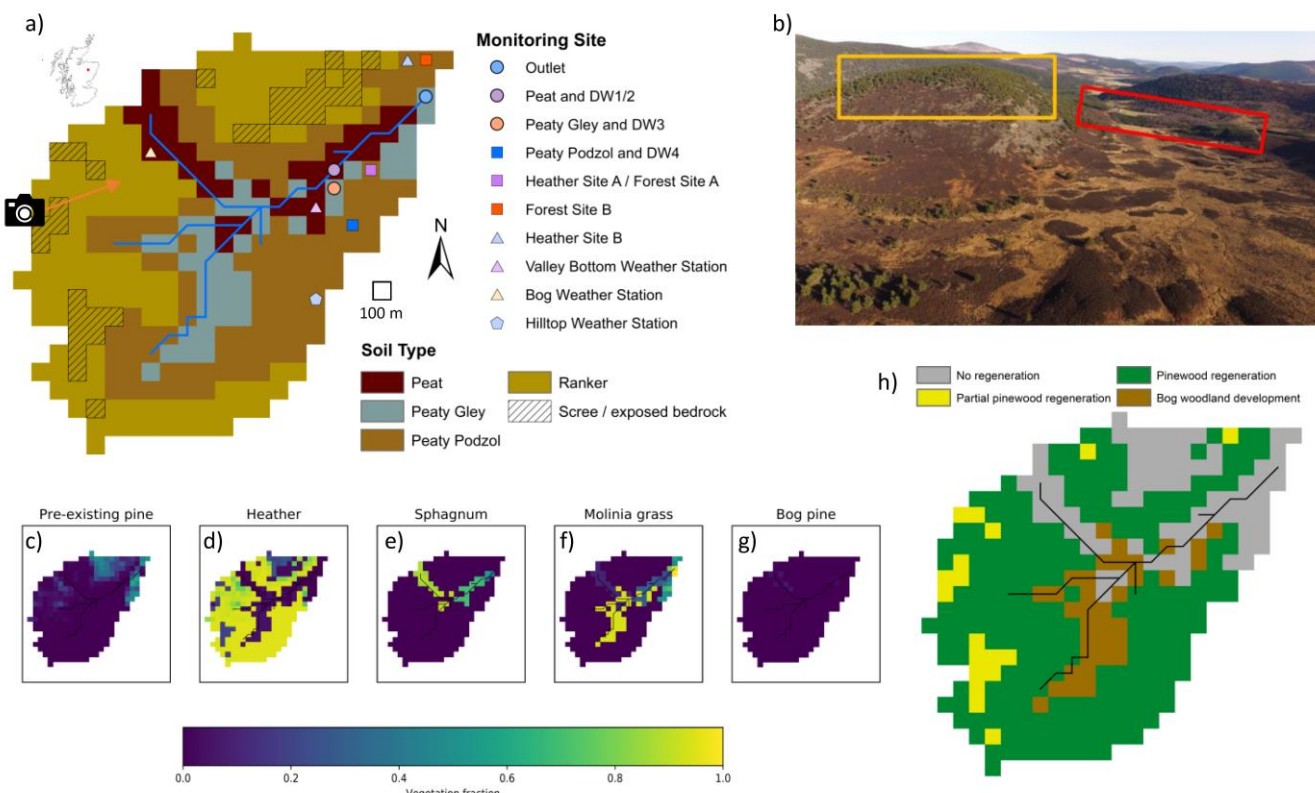

**Figure 1: Characteristics of the Bruntland Burn catchment: a) Map showing distribution of soil types and monitoring sites; b) Aerial view showing current distribution of vegetation types in the catchment - yellow and red boxes show areas of established natural**

**forest and land manged for plantations, respectively, whilst direction of the photo is shown by the arrow in a); c-g) Present-day vegetation coverage in the catchment (as also used in the baseline scenario simulation); h) Map of regeneration potential.**

## 3 Methods

### 3.1 The EcH₂O-iso model

EcH$_2$O-iso is a development of the ecohydrological model EcH$_2$O (Maneta and Silverman, 2013). It consists of three tightly

coupled modules simulating the water balance, vertical energy balance and vegetation growth dynamics, and an additional

fourth module that tracks the stable water isotope composition and ages of hydrological stores and fluxes (Kuppel et al., 2018a).

The model domain is defined by a regularly grided digital elevation model (DEM) that sets local flow directions, and governing

equations are solved for fixed timesteps using finite-differences. Proportional coverage of different vegetation types (based on

physiology and structure) and bare soil are specified for each grid cell.






The energy balance is resolved sequentially for the canopy and soil surface. Canopy temperature is determined iteratively to balance latent (transpiration and evaporation of intercepted water) and sensible heat fluxes with net radiation reaching the canopy. Interception evaporation is limited by available intercepted water whilst a Jarvis-type stomatal conductance model limits transpiration. Transpiration demand is satisfied by root water uptake from three soil layers (L1, L2 and L3) in proportion

to the water content and fraction of roots in each layer. An exponential function determines the latter (Kuppel et al., 2018b). At the soil surface, iteratively determined temperature partitions net radiation and heat advected by rainfall/throughfall into latent heat for snowmelt and soil evaporation from L1, sensible heat exchanges between the soil and atmosphere, and heat into the ground and snowpack. Soil evaporation is limited by the moisture content of L1. In addition to soil, two further hydrological stores are conceptualised: canopy interception storage and ponded water. Once interception storage is full, throughfall reaches

the ponded water store where it may infiltrate into L1 based on the Green-Ampt model. Vertical redistribution of water occurs via gravitational drainage when volumetric water content (VWC) in any of the soil layers exceeds field capacity. Gravitational water in L3 can leak from the model domain or move laterally as GW simulated via a kinematic wave model. Water remaining in ponded storage is routed to the next downslope cell as overland flow that can potentially travel the length of the catchment to reach the stream within one timestep. GW also seeps to the stream, with channel routing simulated using a kinematic wave

model. Stable water isotopes and water ages are tracked assuming complete mixing (Eq. 1 in Smith et al., 2020), with isotopic fractionation due to evaporation in L1 simulated via the Craig-Gordon model (Craig and Gordon, 1965; Kuppel et al., 2018a). Vegetation growth dynamics were not simulated in this application; consequently, vegetation characteristics were static within each considered scenario. Previous work provides further details of $EcH_2O$ and $EcH_2O$-iso (Kuppel et al., 2018a; 2018b; Maneta and Silverman, 2013; Smith et al., 2020).


### 3.2 Present-day baseline scenario

Catchment soil distribution was based on major Hydrology of Soil Types (HOST) classifications (Fig. 1a; Tetzlaff et al., 2007). Soil types were assumed to be spatially uniform. Five vegetation types characterised the present-day baseline scenario: Pre-existing Scots pine, heather (also used to represent other understory shrubs such as bilberry), *Molinia* grass, *Sphagnum* and

bog pine (Table 1). LiDAR-based estimates of canopy cover were used to derive proportional tree coverages in each cell (c.f. Kuppel et al., 2018b); trees on the podzols/rankers and on the wetter peat/peaty gleys were designated as pre-existing pine (Fig. 1c) and bog pine (Fig. 1g), respectively, enabling stunted development of the latter due to waterlogging to be explicitly represented (McHaffie et al., 2002). The extents of remaining vegetation types (Table 1; Fig. 1d-f) were derived from the soil distribution, field mapping and aerial imagery (Kuppel et al., 2018b; Tetzlaff et al., 2007). To account for scree and exposed

bedrock, some rankers on the western and northern hillslopes were set with bare earth coverages of 80% and up to 95% of the treeless surface, respectively (Fig. 1a). All vegetation heights in the baseline scenario were based on local knowledge (Table 1).



**Table 1: The vegetation types considered in the baseline and regeneration scenarios along with their distribution by soil type, proportional aerial coverage, height, and leaf area index (LAI) scaling.**

| Scenario and vegetation type | Soil distribution | Cover | Height (m) | LAI scale factor [a] |
|---|---|---|---|---|
| *Baseline* | | | | |
| Pre-existing pine | Podzol & ranker | LiDAR-derived | 10 | 1 |
| Heather | Peaty gley | 5% of treeless area | 0.4 | - |
| | Podzol & ranker | 95% of treeless area | 0.4 | - |
| | | (5 to 40% of treeless area if scree or bedrock present) | | |
| *Sphagnum* | Peat | 70% of treeless area except in NW (90% of treeless area) | 0.1 | - |
| *Molinia* grass | Peaty gley & peat | 99% of tree-, shrub- and moss-less area | 0.5 | - |
| Bog pine | Peat & peaty gley | LiDAR-derived | 3.4 | 0.17 |
| *Thicket woodland* | | | | |
| Pre-existing pine | Podzol & ranker | As baseline in areas of present-day native pinewood and plantation | 10 | 1 |
| Heather | Peaty gley | 75% of available treeless bog woodland area [b] | 0.4 | - |
| | Podzol & ranker | 9% of available treeless pinewood regeneration area [c] | 0.12 [c] | - |
| | | As baseline in area of present-day native pinewood | 0.4 | - |
| *Sphagnum* | Peat | As baseline | | |
| *Molinia* grass | Peat | As baseline | | |
| | Peaty gley | 99% of available tree- and shrub-less bog woodland area [b] | 0.5 | - |
| Bog pine | Peaty gley | 15% of available bog woodland area [d, e] | 2.4 | 0.04 |
| | Peat | As baseline | | |
| Thicket pine | Podzol & ranker | 95% of available pinewood regeneration area [f] | 12.7 [g] | 1.37 |
| *Old open woodland* | | | | |
| Pre-existing pine | Podzol & ranker | As baseline in areas of present-day native pinewood and plantation | 10 | 1 |
| Heather | Peaty gley | 75% of available treeless bog woodland area [b] | 0.4 | - |





| | Podzol & ranker | 82% of available treeless pinewood regeneration area [c] | 0.29 [c] | - |
| | | As baseline in area of present-day native pinewood | 0.4 | - |
| *Sphagnum* | Peat | As baseline | | |
| *Molinia* grass | Peat | As baseline | | |
| | Peaty gley | 99% of available tree- and shrub-less bog woodland area [b] | 0.5 | - |
| Bog pine | Peaty gley | 15% of available bog woodland area [d, e] | 8.4 | 0.4 |
| | Peat | As baseline | | |
| Old open pine | Podzol & ranker | 54% of available pinewood regeneration area [f] | 15.5 [g] | 0.59 |

Notes: [a] With respect to calibrated Pre-existing pine leaf area index (*LAI*); [b] Steven and Carlisle (1959); [c] Parlane et al. (2006); [d] McHaffie et al. (2002); [e] Summers (2018); [f] Summers et al. (1997); [g] Summers et al. (2008).

### 3.3 Baseline calibration

The baseline scenario was simulated from 21 February 2013 to 08 August 2016 on a 100×100 m grid with a daily timestep. Model forcing data are detailed in Table S1. Six years of looped data were used to spin-up the model for calibration whilst 30 years were used for post-calibration runs to stabilise water ages (c.f. Kuppel et al., 2020). Calibration followed a Morris
sensitivity analysis (Morris, 1991; Sohier et al., 2014) to identify sensitive parameters (Table S2). For efficiency, it was assumed that pre-existing pine and bog pine vegetation types could take the same parameter values except for *leaf area index (LAI)*; thus, only four sets of vegetation parameters required calibration. Overall, 90 parameters (10 × 4 for soil, 12 × 4 for vegetation and 2 for channel) were calibrated. The parameter space was sampled by conducting 100,000 Monte Carlo simulations. Parameter values were drawn from initial ranges informed by Kuppel et al. (2018b) and additional literature
reviews (Table S2). The *LAI* of bog pine was related to the sampled *LAI* of pre-existing pine via a scale factor accounting for relative differences in canopy architecture. The scale factor (0.17) was the proportional difference between an empirical estimate of LAI for bog pine and a local measurement of LAI for Scots pine (Wang et al., 2018). The former was obtained by first estimating below-canopy irradiance as a function of tree height (3.4 m) and density (275 trees ha$^{-1}$; Summers et al., 1997) via the equation of Parlane et al. (2006). This irradiance was then used in Beer's Law with a light-extinction coefficient of 0.5
(White et al., 2000) to determine LAI.

Diverse ecohydrological and isotope datasets were available at the BB for model calibration (Table 2). Protocols used to collect and process these data are detailed in Kuppel et al. (2018a; 2018b). In most cases, model outputs were directly compared against relevant observed datasets. Simulated soil variables (VWC and bulk water isotopes) were exceptions; these are output
for each soil layer and, therefore, do not directly correspond to depth-specific observations (c.f. Beven, 2006). To accommodate





this, simulated VWC timeseries for L1 and L2 were compared to observations made at the depth closest to the mid-point of each layer. Meanwhile, simulated soil water isotopes in L1 and L2 were compared with the depth-averaged isotopic composition of the soil profile encompassed by each respective layer. Consequently, observations compared to simulated outputs from L1 and L2 could vary depending on soil depth parameterisation.


Model skill in simulating dynamics of ecohydrological and isotopic observations was quantified using the performance metrics in Table 2. Mean absolute error (MAE) was used for discharge to avoid over-emphasising high flows (Legates and McCabe, 1999), and for all isotope simulations given limited observational variability and daily timestep of the model (c.f. Gupta et al., 2009; Schaefeli and Gupta, 2007). Root mean squared error (RMSE) was otherwise used as this is recommended if no

information is available on error distributions (Chai and Draxler, 2004; Kuppel et al., 2018b). To determine behavioural parameter sets, model runs were first selected that simulated saturation areas <60% of total catchment area for at least 90% of the simulation period; this reflected mapping and modelling of the extent of saturation in the BB (Birkel et al., 2010). Performance metrics for each calibration dataset were then ranked across all runs that satisfied this constraint. Runs were finally ordered by their worst-performing metric with the "best" 30 runs being retained as behavioural.


**Table 2: Datasets used in calibration of EcH₂O-iso. The performance metrics used to quantify model skill in simulating each dataset and the ranges in values achieved by behavioural model runs are also given.**

| Dataset | Temporal coverage | Metric [a] | Behavioural range |
|---|---|---|---|
| Streamflow | Full | MAE | 0.026 to 0.033 $m^3 s^{-1}$ |
| *Soil moisture content* | | | |
| Forest A (10, 20 & 40 cm) | Full | RMSE | [b] 0.10 to 0.27 $m^3 m^{-3}$ |
| Forest B (10, 20 & 40 cm) | 25 February 2015 onwards | RMSE | [b] 0.11 to 0.19 $m^3 m^{-3}$ |
| Gley (10, 30 & 50 cm) | Full | RMSE | [b] 0.04 to 0.29 $m^3 m^{-3}$ |
| Heather A (10, 20 & 40 cm) | Full | RMSE | [b] 0.06 to 0.27 $m^3 m^{-3}$ |
| Peat (10 cm) | Full | RMSE | 0.02 to 0.10 $m^3 m^{-3}$ |
| Peaty podzol (10, 30 & 50 cm) | Full | RMSE | [b] 0.03 to 0.11 $m^3 m^{-3}$ |
| *Green fluxes* | | | |
| Heather A: transpiration and evapotranspiration | 31 July 2015 to 30 October 2015 & 21 April 2016 to 03 August 2016 | RMSE | T: 0.50 to 0.69 $mm\ d^{-1}$ ET: 0.81 to 1.16 $mm\ d^{-1}$ |
| Heather B: transpiration and evapotranspiration | 31 July 2015 to 30 October 2015 & 31 March 2016 to 11 July 2016 | RMSE | T: 0.43 to 0.60 $mm\ d^{-1}$ ET: 0.78 to 0.95 $mm\ d^{-1}$ |





| | | | |
|---|---|---|---|
| Forest A: transpiration | 08 July 2015 to 27 September 2015 | RMSE | 0.33 to 0.70 mm d$^{-1}$ |
| Forest B: transpiration | 01 April 2016 onwards | RMSE | 0.22 to 0.39 mm d$^{-1}$ |
| *Net radiation* | | | |
| Bog weather station | 10 October 2014 to 03 August 2016 | RMSE | 29 to 36 W m$^{-2}$ |
| Hilltop weather station | 22 April 2015 onwards | RMSE | 40 to 58 W m$^{-2}$ |
| Valley bottom weather station | 10 October 2014 onwards | RMSE | 29 to 39 W m$^{-2}$ |
| Streamflow: $\delta^2$H | Full | MAE | 2.7 to 8.2‰ |
| *Soil $\delta^2$H (2.5, 7.5, 12.5 and 17.5 cm)* | | | |
| Forest A: bulk soil water | 29 September 2015 onwards (monthly) | MAE | [b] 3.4 to 20.6‰ |
| Forest B: bulk soil water | 29 September 2015 onwards (monthly) | MAE | [b] 3.8 to 7.5‰ |
| Heather A: bulk soil water | 29 September 2015 onwards (monthly) | MAE | [b] 2.6 to 21.5‰ |
| Heather B: bulk soil water | 29 September 2015 onwards (monthly) | MAE | [b] 4.2 to 8.2‰ |
| *Groundwater $\delta^2$H* | | | |
| Deeper well 1 (Peat) | 11 samples between 09 June 2015 and 22 July 2016 | MAE | 0.4 to 5.4‰ |
| Deeper well 2 (Peat) | 11 samples between 09 June 2015 and 22 July 2016 | MAE | 0.7 to 2.9‰ |
| Deeper well 3 (Peaty gley) | 11 samples between 09 June 2015 and 22 July 2016 | MAE | 0.7 to 6.7‰ |
| Deeper well 4 (Peaty podzol) | 11 samples between 09 June 2015 and 22 July 2016 | MAE | 0.7 to 3.4‰ |

Notes: [a] MAE = mean absolute error, RMSE = root mean squared error; [b] Range across first two soil layers of EcH$_2$O-iso.

### 3.4 Regeneration scenarios

Extensive work characterising the structure of native pine stands at Abernethy Forest in the Cairngorms National Park (Parlane
et al., 2006; Summers, 2018; Summers et al., 1997; 2008) was used to select two stages of forest regeneration for simulation.
For context, Fig. S1 summarises the general sequence of natural pinewood regeneration. The thicket stage has the highest tree
densities and near-complete canopy closure; consequently, this stage was selected because it likely has the most substantial
impact on water partitioning and catchment hydrology. Trees are ~40 years old whilst understorey development is limited. The
second chosen stage was old open forest as a possible realisation of old-growth forest. This stage may be somewhat semi-
natural and, thus, characterised by a more open canopy than might be expected (reflecting limited modification by grazing
pressures and selective felling; Summers et al., 2008). However, it was assumed to represent the near-end point of a rewilding
trajectory that balances human demands on the land with forest regeneration (c.f. Deary and Warren, 2017), and a possible





"lower impact" stage of forest development. Trees are tall, old (~150 years) and sparsely distributed with an understory of well-developed shrubs.


The proportional coverage and characteristics of vegetation at each stage of forest regeneration are given in Table 1. Native pinewoods, consisting of thicket/old-open pine and a heather understory, were assumed to fully regenerate on podzols and rankers (Fig. 1h), reflecting the preference of pine for freely draining minerogenic soils (Mason et al., 2004). The available regeneration area was limited in ranker cells containing scree or exposed bedrock such that the bare earth fraction remained

constant across scenarios, whilst regeneration did not occur on managed land at the catchment outlet or in pre-existing areas of native pinewood on the northern hillslope (Fig. 1h). Bog woodland consisting of stunted bog pine, heather and *Molinia* grass, was simulated to develop on the wetter peaty gleys (McHaffie et al., 2002; Summers et al., 2008), whilst no regeneration was possible on the waterlogged peat (Fig. 1h). Spatial changes in vegetation cover for each regeneration scenario relative to the baseline are shown in Fig. 2. Scale factors relating calibrated values of *LAI* for pre-existing pine to *LAI* of thicket, old-open

and bog pine were calculated as described in Sect. 3.3. For thicket/old-open pine, measured below-canopy irradiance was available from Parlane et al. (2006) and, hence, did not require estimation. For bog pine, irradiance was again obtained via the equation of Parlane et al. (2006). The heights of bog pine in each scenario were estimated by first calculating a "stunted" growth rate (~0.06 m yr$^{-1}$) by dividing the height of present-day bog pine by an assumed age of 60 years. This was multiplied by the age of pines in the thicket (40 years) and old open (150 years) forests to give bog pine heights of 2.4 m and 8.4 m for

each scenario, respectively. These values are broadly consistent with height measurements made by Summers et al. (2008) on bog pine trees with an interquartile age range between 72 and 143 years.

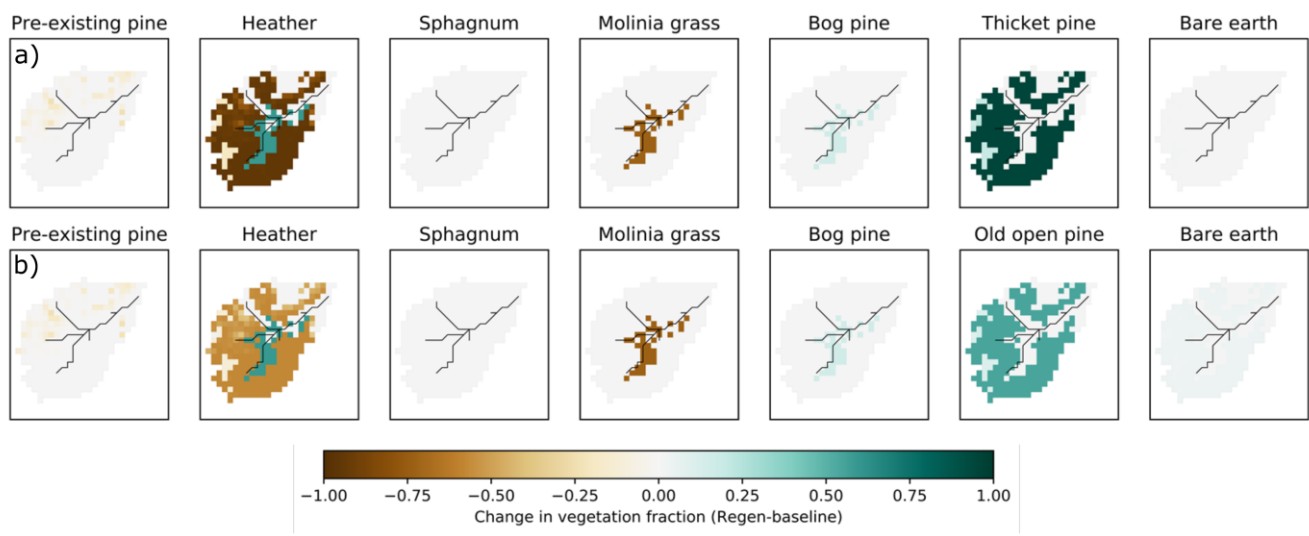

**Figure 2: Changes in vegetation cover for the a) Thicket scenario and b) Old open forest scenario. Changes are reported as**
**regeneration scenario minus baseline scenario.**



Simulations were driven by the 30 behavioural parameter sets obtained from the baseline calibration and undertaken for the same time period (21 February 2013 to 08 August 2016) with 30 years of spin-up. As previously stated, soil properties remained unchanged in the regeneration scenarios. Additionally, the vertical root distribution parameter was consistent amongst pine

vegetation types as after the sapling stage, the most significant changes to Scots pine rooting characteristics are expressed in the horizontal rather than vertical direction (Laitakari, 1927; Makkonen and Helmisaari, 2001). Potential effects of such changes on the vertical root distribution would likely already be captured by the sampling range of this parameter for pre-existing pine (Table S2).

**3.5 Change analysis**

To contextualise changes to water flux partitioning, simulated maximum root zone (RZ) and interception storage capacities were quantified for each scenario. For each vegetation type in a given grid cell, RZ storage capacity was defined as the sum of maximum plant available VWC in each layer weighted by root fraction, multiplied by the coverage-weighted average rooting depth of all vegetation types in the cell. A coverage-weighted sum across all present vegetation types then yielded the total RZ

storage capacity for the cell. The total interception storage capacity of each cell was similarly calculated, with the interception storage capacity of each vegetation type obtained by multiplying the parameters *LAI* and *maximum canopy storage* (Table S2). Average values of RZ and interception storage capacity for the catchment were then calculated.

Catchment-scale flux partitioning was assessed by quantifying seasonally averaged totals of simulated discharge at the outlet,

and catchment-averaged GW recharge, soil and interception evaporation, and transpiration fluxes. Seasons were defined as April to September and October to March, broadly corresponding to periods of biological growth and dormancy, respectively (Dawson et al., 2008). Seasonally averaged water ages were also calculated for selected stores and fluxes. Daily timeseries of discharge, stream isotopic composition and water age provided a spatially integrated insight into how regeneration affected modelled catchment hydrology at a higher temporal resolution. To better understand spatial drivers of changes to catchment-

scale flux partitioning, differences in seasonal daily average "blue" and "green" fluxes between each scenario and the baseline were spatially disaggregated.

To assess how regeneration impacts hydrological source areas and runoff generation, the spatial extent of hydrological connectivity was quantified under contrasting flow conditions. A cell was considered hydrologically connected if overland

flow (OLF) was simulated for the cell and all downslope cells along the flow path to the stream. Flow path lengths for connected cells were calculated by accumulating the straight line or diagonal lengths (dependent on local flow direction) of all cells along the flow path (c.f. Turnbull and Wainwright, 2019). A threshold of OLF was not implemented as water can reinfiltrate along a flow path in EcH₂O-iso (Maneta and Silverman, 2013). Three dates were selected for connectivity analysis.



The first, 22 July 2013, was during a particularly dry early summer. The second, 10 August 2014, was during a moderately
wet late summer period caused by a frontal storm. The third, 30 December 2015, was at the height of some of the worst
flooding seen for 100 years in NE Scotland caused by Storm Frank (Marsh et al., 2016).

# 4 Results

## 4.1 Baseline calibration

Figure 3 summarises model skill in simulating "blue" and "green" fluxes, isotope dynamics, and net radiation at selected
monitoring locations (remaining simulations shown in Figs. S2-5). Calibrated parameter ranges and performance metrics are
given in Tables S2 and 2, respectively. Stream discharge was generally well simulated, with only occasional under-prediction
of baseflows (e.g. summer 2016) and the most extreme events (Fig. 3b). At most sites, modelled VWC in L1 and L2 was
bracketed by the range observed across the soil profile (Figs. 3c and S2), although simulations were sometimes less dynamic
than observations and RMSEs could be large (Table 2). However, this likely reflects the commensurability issues highlighted
in Sect. 3.3 (also relevant for soil water isotopes), and the fact that Heather Site A and Forest Site A fell within the same model
cell. The model was generally successful in reproducing dynamics of stream, soil water and groundwater isotopes (Table 2;
Figs. 3d-f and S3), implying internal catchment functioning was reasonably well-captured. Stream isotopes were sometimes
over-enriched, suggesting slightly high soil evaporation (Fig. 3d); however, the variability was generally well-captured. Larger
deviations during events likely reflect structural limitations of the model (e.g. the ability of OLF to traverse the catchment
within one timestep). Excessive evaporation was not apparent from simulated soil water isotopes (Figs. 3e and S3), although
averaging over L1 and L2 could obscure the effects of evaporative fractionation in the former. The model had skill in simulating
ET and forest transpiration (Table 2; Figs. 3g-h and S4). However, underestimation of transpiration for the heather (Fig. S4)
may indicate too much radiative energy being used for evaporation. Seasonality in net radiation was well simulated; however,
the magnitude of shorter-term variability was under-estimated in summer (Table 2; Figs. 3i and S5).


## 4.2 Impact of regeneration on water storage capacity

Figure 4 summarises simulated RZ and interception storage capacities for each scenario. Median RZ storage capacity increased
by 21 mm in the thicket forest scenario (Fig. 4a) reflecting replacement of heather by thicket pine (Fig. 2a) with deeper roots
(Table S2) that increase access to water stored in L2 and L3. Small increases in RZ storage capacity were simulated for the
old open forest scenario, albeit with greater overlap with the baseline (Fig. 4a), likely reflecting the more balanced composition
between pine and heather (Fig. 2b). This, along with greater proportional coverage of bare earth (Fig. 2b), may also explain
overlap of interception storage capacity between the old open forest and baseline scenarios (Fig. 4b). The greater coverage of
thicket pine (Fig. 2a) with a dense canopy substantially increased interception storage capacity for the thicket forest scenario
(Fig. 4b).





**Figure 3: Time series of a) Precipitation; and of observed and simulated b) Discharge; c) Volumetric water content (VWC) at the peaty podzol site; d) Stream δ²H composition; e) Bulk soil water (SW) δ²H under Forest B; f) Groundwater δ²H at Deeper Well 2 (DW2); g) Total evapotranspiration (ET) for Heather A; h) Transpiration (Tr) for Forest B; i) Net radiation (CNR) at the valley bottom weather station. 90% spread of simulations are from the 30 behavioural model runs.**





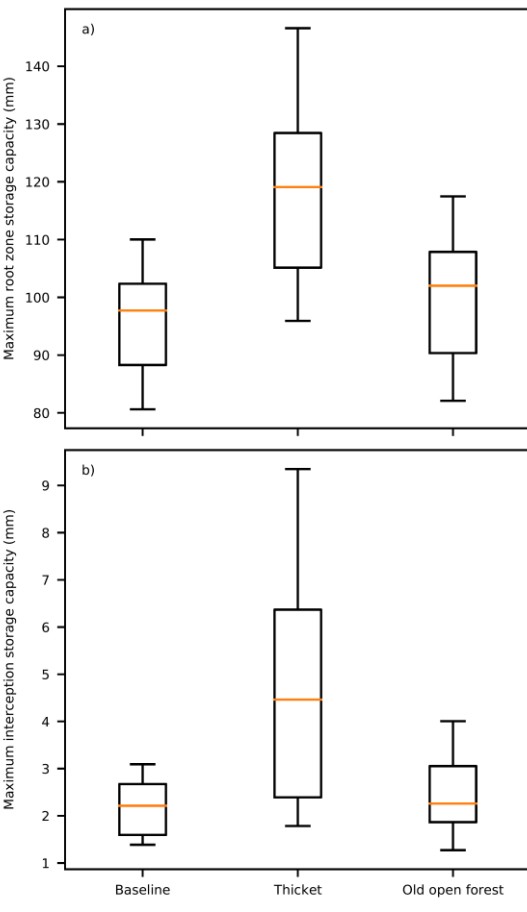

**Figure 4: Boxplots showing maximum a) Root zone storage capacity and b) Interception storage capacity, for the baseline and forest regeneration scenarios. The median is shown by the orange line whilst the box extends from the lower to upper quartiles. The 5th and 95th percentiles are given by the tails.**


### 4.3 Changes to catchment-scale water flux partitioning

Figure 5 shows how impacts of modelled regeneration integrated to affect the simulated quantity and isotopic composition of streamflow. The greatest differences in discharge between the thicket and baseline scenarios were for moderate to low flow periods with the magnitudes of high flows being relatively consistent (Fig. 5b). For the old open forest scenario, discharge was
similar to the baseline. For both regeneration scenarios, simulated stream isotope dynamics were comparable to the baseline (Fig. 5c); however, streamwater could sometimes be slightly more depleted in summer for the thicket scenario indicating reduced soil evaporation.



**Figure 5: Timeseries of a) Observed precipitation; and simulated b) Discharge; c) Stream δ²H composition; d) Streamwater age. In b) to d), the area between the two black lines and the areas that are shaded represent the 90% spread of simulations from the 30 behavioural model runs.**

Overall, behavioural models consistently simulated a decrease in seasonally averaged discharge for the thicket scenario (Table 3a-b). This resulted from increased interception evaporation (Table 3e-f) and transpiration (Table 3i-j), and decreased GW recharge (Table 3c-d). Recharge was most reduced for Oct-Mar, concurrent with the greatest increase in interception evaporation. Soil evaporation was lower for the thicket scenario (Table 3g-h), likely reflecting limits imposed by lower soil





moisture due to greater interception evaporation and transpiration losses, and reduced penetration of radiation through the thicket canopy. Differences in seasonally averaged fluxes were much smaller between the old open forest and baseline (Table
3b, d, f, h, j), resulting in the median and 5th/95th percentile seasonally averaged flux totals generally being similar for the two scenarios (Table 3a, c, e, g, i). However, there was a repartitioning of water between increased interception evaporation and reduced GW recharge for Oct-Mar that contributed to a ~30 mm decrease in simulated median total discharge (Table 3a).

**Table 3: Seasonally averaged water flux totals and differences in seasonally averaged totals between the regeneration and baseline**
**scenarios. Totals and differences were calculated for each behavioural run individually and summarised by the median and 5th/95th percentile (in square brackets) values across all behavioural runs. Differences are reported as regeneration scenario minus baseline.**

| | Median [5th/95th percentile] seasonally averaged flux totals (mm over summary period) | | Median [5th/95th percentile] differences in seasonally averaged flux totals (mm over summary period) | |
|---|---|---|---|---|
| | Apr to Sep | Oct to Mar | Apr to Sep | Oct to Mar |
| *Outlet stream discharge* | a) | | b) | |
| Baseline | 195.9 [158.8, 231.6] | 449.2 [414.7, 480.5] | - | - |
| Thicket | 143.3 [97.2, 201.7] | 316.0 [258.4, 434.6] | -63.8 [-97.3, -7.8] | -128.7 [-178.0, -26.4] |
| Old open forest | 189.5 [158.6, 229.6] | 416.2 [369.1, 486.4] | -12.0 [-31.1, 18.0] | -28.7 [-59.3, 15.0] |
| *GW Recharge* | c) | | d) | |
| Baseline | 161.3 [140.0, 190.1] | 352.7 [282.9, 402.0] | - | - |
| Thicket | 107.0 [68.1, 156.1] | 277.0 [223.7, 348.7] | -62.6 [-92.7, -5.5] | -80.2 [-109.0, -15.4] |
| Old open forest | 153.0 [128.5, 185.2] | 335.9 [269.4, 381.5] | -9.3 [-26.7, 19.1] | -17.6 [-36.5, 6.5] |
| *Interception evaporation* | e) | | f) | |
| Baseline | 102.2 [80.8, 131.4] | 76.9 [61.6, 90.2] | - | - |
| Thicket | 182.5 [103.8, 234.2] | 196.3 [102.2, 253.0] | 82.6 [-3.0, 123.6] | 118.5 [26.7, 167.6] |
| Old open forest | 115.8 [73.8, 158.9] | 105.6 [60.9, 150.0] | 19.4 [-24.4, 42.6] | 31.3 [-9.8, 64.3] |
| *Soil evaporation* | g) | | h) | |
| Baseline | 74.2 [52.3, 93.5] | 41.2 [35.0, 44.6] | - | - |





| | | | | |
|---|---|---|---|---|
| Thicket | 44.0 [34.0, 57.1] | 28.2 [22.1, 33.6] | -32.3 [-47.3, -7.6] | -12.4 [-16.9, -6.9] |
| Old open forest | 72.0 [55.5, 90.5] | 37.5 [32.5, 40.8] | -1.7 [-12.4, 13.0] | -3.7 [-5.9, 0.1] |
| *Transpiration* | **i)** | | **j)** | |
| Baseline | 66.4 [51.7, 79.9] | 6.1 [4.3, 7.7] | - | - |
| Thicket | 89.4 [71.5, 120.7] | 12.1 [8.6, 19.8] | 27.1 [11.2, 47.4] | 6.1 [2.8, 12.8] |
| Old open forest | 57.6 [45.4, 75.5] | 6.1 [4.6, 8.4] | -6.8 [-13.7, 2.9] | 0.0 [-1.0, 1.7] |

## 4.4 Spatio-temporal dynamics of baseline flux partitioning

Figure 6 summarises spatial dynamics of median seasonal daily average "blue" and "green" fluxes for the baseline scenario.
Simulated OLF was more limited for Apr-Sept (Fig. 6a). The largest fluxes were simulated for the peats and peaty gleys, with some OLF also being generated by the ranker soils on the hillslopes. The latter is interpreted as representing the rapid near-surface flows in the shallow rankers that are driven by emergence of bedrock fracture flow at slope breaks. OLF fluxes were greater and had similar spatial patterns for Oct-Mar with additional fluxes generated from the podzols on lower hillslopes in the north and south (Fig. 6a). Vertical movement of water (infiltration and GW recharge) was greatest in winter and mostly
occurred on the podzols; the largest fluxes were simulated at the boundaries between the podzols and rankers (Fig. 6b and d) reflecting lateral flows from upslope moving vertically in deeper soils. Water was then simulated to move downslope as GW (Fig. 6e) to sustain saturation in the valley bottom, as evidenced by high exfiltration fluxes especially in Oct-Mar (Fig. 6c).

Daily average fluxes of ET were simulated as greatest for Apr-Sept, particularly in the valley bottom (Fig. 6f). This was
facilitated by the wet peat and peaty gleys maintaining transpiration (Fig. 6g) and soil evaporation (Fig. 6h), and further by evaporation of water intercepted by the *Sphagnum* "canopy" (Fig. 6i and Table S2). Dominance of vertical sub-surface fluxes (Fig. 6b and d) limited transpiration (Fig. 6g) and, particularly, soil evaporation (Fig. 6h) from the podzols, reducing total ET fluxes (Fig. 6f). For Oct-Mar, spatial patterns in total ET were less distinct (Fig. 6f) reflecting reduced soil and interception evaporation in the valley bottom (Fig. 6h-i) and the fact that daily transpiration fluxes were essentially 0 (Fig. 6g). A notable
pattern was that total ET was greatest where there was substantial pre-existing pine (Figs. 1 and 6f) due to sustained interception evaporation (Fig. 6i).





**Figure 6: Maps showing median daily average fluxes in April to September and October to March for: a) Overland flow; b) Infiltration into L1 of the soil; c) Exfiltration to the surface; d) Groundwater (GW) recharge; e) Groundwater flow, f) Total evapotranspiration (ET); g) Transpiration; h) Soil evaporation; and i) Interception evaporation.**



### 4.5 Spatio-temporal disaggregation of regeneration effects on flux partitioning

Median differences in seasonal daily average "blue" fluxes were most dramatic for the thicket scenario (Fig. 7). For both Apr-
Sept and Oct-Mar, similar spatial patterns were simulated, although differences tended to be greater for the latter. More limited
OLF generation by the rankers (Fig. 7a) led to similar-magnitude reductions in daily average infiltration and GW recharge on
the podzols (Fig. 7b and d). Consequently, downslope movement of GW also decreased in both Apr-Sept and Oct-Mar (Fig.
7e). This contributed to a drying out of the valley bottom (especially in Oct-Mar) even though local regeneration was limited
(Fig. 2a). Daily average OLF fluxes were simulated to decrease around the fringes of the stream for both Apr-Sept and Oct-
Mar (Fig. 7a). The largest decreases occurred in the NW of the catchment as a direct consequence of reduced OLF from
upslope. Elsewhere, OLF reductions strongly reflected the reductions in upslope GW subsidies as evidenced by consistently
decreased exfiltration fluxes (Fig. 7c) and increased infiltration of incoming precipitation in the valley bottom for Oct-Mar
(Fig. 7b) to replenish drier soil and GW stores (Fig. 7d).

Similar spatial dynamics were simulated for the old open forest scenario, however median differences in seasonal daily average
fluxes were much reduced (Fig. 7). It is also noteworthy that even in Oct-Mar, the valley bottom dried out less than in the
thicket scenario; daily average GW flows through the podzols were only simulated to decrease by  <1.5 mm d$^{-1}$ (Fig. 7e),
whilst no increases in infiltration or GW recharge were simulated in the valley bottom (Fig. 7b and d).

Differences in seasonal daily average "green" fluxes were also more apparent in the thicket scenario (Fig. 8). Daily average
ET from the podzols and rankers was simulated to increase throughout the year (Fig. 8a). For Apr-Sept, this resulted from
greater transpiration (Fig. 8b) and, predominantly, interception evaporation (Fig. 8d), reflecting the increased coverage of
thicket pine (Fig. 2a). Reduced penetration of radiative energy through the thicket canopy, and limits imposed by greater water
losses to transpiration and interception evaporation, decreased simulated daily average soil evaporation (Fig. 8c). For Oct-Mar,
increased ET was more driven by greater interception evaporation (Fig. 8d). ET from the bog woodland was similar to the
baseline (Fig. 8a). This was due to decreased transpiration from reduced cover of potentially deep-rooted (Aerts et al., 1992;
Taylor et al., 2001) *Molinia* grass (Figs. 2a and 6b) being compensated by increases in soil (Fig. 8c) and, particularly,
interception evaporation (Fig. 8d). Daily average ET for Apr-Sept decreased in some areas of peat in the NW of the catchment,
despite no regeneration having taken place (Fig. 8a). This would be consistent with drying of the valley bottom limiting summer
transpiration (Fig. 8b). Given small transpiration demands in Oct-Nov (Fig. 6g) no reduction in ET was simulated.





**Figure 7: Maps showing the median difference in daily average fluxes for April to September and October to March: a) Overland flow; b) Infiltration into L1 of the soil; c) Exfiltration to the surface; d) Groundwater (GW) recharge; e) Groundwater flow. Differences are reported as regeneration scenario minus baseline scenario.**





**Figure 8: Maps showing the median difference in daily average fluxes for April to September and October to March: a) Total evapotranspiration (ET); b) Transpiration; c) Soil evaporation; and d) Interception evaporation. Differences are reported as regeneration scenario minus baseline scenario.**





Differences in simulated ET were much more subdued for the old open forest scenario (Fig. 8a). For Apr-Sept, soil evaporation on the podzols and rankers remained similar to the baseline (Fig. 8c) whilst transpiration decreased (Fig. 8b), likely reflecting greater bare earth coverage (Fig. 2b). This offset greater losses to interception evaporation (Fig. 8d) so that only very small

increases in total ET were simulated. Small ET decreases mostly reflected replacement of larger clusters of pre-existing pine with old open forest (Fig. 2b) reducing transpiration (Fig. 8b). A larger increase in interception evaporation from the old open forest (Fig. 8d) drove the increases in ET from the podzols and rankers for Oct-Mar (Fig. 8a).

## 4.6 Effects of regeneration on ages of "blue" and "green" water fluxes

Streamflow, lateral GW outflows and soil water/evaporation were near-consistently simulated to have older average ages in the thicket scenario relative to the baseline (Table 4); however, change magnitudes were often less than the width of simulation uncertainty bounds leading to overlap in flux ages. Simulated daily dynamics revealed that streamwater ages for the thicket scenario could be much older for low to moderate flows, although uncertainty bounds were again wide (Fig. 5d). Relatively young streamwater ages persisted in larger events. Transpiration fluxes were the only ones to indicate a possible slight

preference for younger water through a reduction in 95th percentile average ages (Table 4i-j). For old open forest, age characteristics were generally restored to those simulated for the baseline (Table 4).

**Table 4: Seasonally averaged water flux ages and differences in seasonally averaged ages. Averages and differences were calculated for each behavioural run individually and summarised by the median and 5th/95th percentile (in square brackets) values across all**
**behavioural runs. Differences are reported as regeneration scenario minus baseline.**

| | Median [5th/95th percentile] seasonally averaged water age (days) | | Median [5th/95th percentile] difference in seasonally averaged water age (days) | |
|---|---|---|---|---|
| | Apr to Sep | Oct to Mar | Apr to Sep | Oct to Mar |
| *Outlet stream discharge* | a) | | b) | |
| Baseline | 647 [355, 989] | 484 [292, 758] | - | - |
| Thicket | 763 [439, 1527] | 610 [373, 1136] | 164 [19, 613] | 128 [22, 440] |
| Old open forest | 643 [367, 1103] | 497 [298, 854] | 30 [-29, 141] | 24 [-17, 108] |
| *GW Outflow* | c) | | d) | |
| Baseline | 521 [375, 774] | 441 [305, 672] | - | - |





| | | | | |
|---|---|---|---|---|
| Thicket | 690 [469, 1202] | 590 [410, 999] | 196 [18, 476] | 163 [20, 405] |
| Old open forest | 545 [397, 824] | 465 [337, 736] | 28 [-30, 99] | 26 [-22, 89] |
| *Soil L1* | **e)** | | **f)** | |
| Baseline | 266 [166, 556] | 293 [173, 565] | - | - |
| Thicket | 301 [192, 696] | 288 [183, 636] | 35 [-8, 193] | 8 [-11, 114] |
| Old open forest | 271 [174, 565] | 291 [179, 581] | 6 [-9, 43] | 5 [-6, 32] |
| *Soil evaporation* | **g)** | | **h)** | |
| Baseline | 276 [168, 673] | 281 [181, 586] | - | - |
| Thicket | 395 [253, 1039] | 329 [218, 823] | 124 [12, 502] | 62 [-4, 349] |
| Old open forest | 291 [189, 683] | 299 [193, 614] | 12 [-42, 96] | 12 [-9, 71] |
| *Transpiration* | **i)** | | **j)** | |
| Baseline | 353 [206, 737] | 365 [227, 780] | - | - |
| Thicket | 361 [204, 642] | 309 [208, 450] | 11 [-106, 71] | -54 [-336, 10] |
| Old open forest | 409 [219, 780] | 414 [237, 727] | 41 [-14, 89] | 26 [-103, 83] |

## 4.7 Changes to hydrological connectivity

Figure 9 summarises hydrological connectivity dynamics for the three investigated dates. Histograms show the number of cells
with given flow path lengths that were connected to the stream for each scenario, whilst maps show the proportion of
behavioural model runs for which a cell was simulated as connected. On 22 July 2013, baseline connectivity was generally
only established for a limited (~5% of potential) number of cells close to the stream (Fig. 9a-b), reflecting the dry summer
conditions. In the thicket scenario, the spatial extent of connectivity became even more limited, though saturation in the valley
bottom was maintained (Fig. 9a-b). Regeneration of old open forest did not have a substantial effect on connectivity dynamics.






**Figure 9: For the end of a prolonged summer dry period (22 July 2013), a) A histogram showing the number of cells with different flow path lengths that were connected to the stream via overland flow, and b) A map showing the proportion of behavioural runs for which each cell was connected to the stream via overland flow. c-d) Are the same for the height of a late summer wet period (10 August 2014) whilst e-f) are for a 100-year return period flood (30 December 2015). Dashed lines in the histograms indicate the total number of cells in the catchment with a given flow path length to the stream.**

The catchment had a much wetter baseline state on 10 August 2014 with, on average, 26% of cells connected to the stream. Connectivity was concentrated in the west of the catchment and along isolated segments of the north and south hillslopes (Fig. 9d). Connected cells generally had flow path lengths of up to 600 m, although more distant cells could also be connected (Fig. 9c). In the thicket scenario, the proportional decrease in the median connectivity of cells with flow paths lengths between 400 and 600 m was less than for cells closer to or further from the stream (Fig. 9c). This reflected persistence of longer connected flow paths predominantly in the west of the catchment but also on the northern hillslope where there is an ephemeral GW seepage track (Fig. 9d; Scheliga et al., 2019). However, the likelihood of cells in the west and also south of the catchment being connected in a behavioural model run did decline (Fig. 9d). In the old open forest scenario, the distribution of connected cells was again very similar to the baseline.

Even greater baseline connectivity was simulated for 30 December 2015 with, on average, 46% of cells connected to the stream (Fig. 9e). The more consistent establishment of connectivity on the northern hillslope and in the SW headwater, increased the number of connected cells with moderate to long (>400 m) flow path lengths (Fig. 9e-f). The podzols in the south remained least connected, though the connected area did extend further upslope (Fig. 9f). In the thicket scenario, spatial patterns of connectivity were similar overall and cells with longer flow paths could still be connected (Fig. 9e-f). However, the median number of connected cells decreased for all flow path lengths (Fig. 9e), with reductions in connectivity most notable in the SW (Fig. 9f). There were also areas where reduced OLF generation from riparian cells substantially limited the connectivity potential of upslope cells (e.g. around the headwater confluences; Figs. 7a and 9f). Connectivity in the old open forest scenario was again comparable to the baseline.

## 5 Discussion

### 5.1 The effect of natural forest regeneration on "blue" and "green" water partitioning

Previous studies investigating the hydrological consequences of changes in forest cover have often sought to understand how conversion and management of land for commercial forestry affects aggregated metrics of catchment hydrological functioning (Ellison et al., 2017; Filoso et al., 2017), especially in the UK context (Marc and Robinson, 2007). Consequently, findings may not be transferable to the case of passively managed natural forest regeneration that is the goal of rewilding efforts in degraded landscapes such as the Scottish Highlands (zu Ermgassen et al., 2018). Therefore, using the EcH$_2$O-iso model, we investigated the ecohydrological consequences of natural pinewood regeneration for the BB catchment in Scotland by comparing simulated



present-day baseline conditions with two land cover change scenarios representing different stages of forest regeneration
(thicket and old open forest). The overall skill of the model in simulating diverse ecohydrological and isotope datasets (Table
2; Fig. 3), its past independent validation in the BB using isotope data (Kuppel et al., 2018a), and consistency of simulated
"blue" fluxes with conceptual models of the BB derived from past empirical (Tetzlaff et al., 2014) and independent modelling
(Ala-aho et al., 2017) studies, gave confidence in using EcH$_2$O-iso for simulating the effects of forest regeneration.

Our major finding was that dynamics of water partitioning deviated most strongly from the baseline during early stages of
regeneration with likely recovery as the forest matured and opened out. During the thicket stage, simulated increases in
interception evaporation principally drove changes to water partitioning (Table 3). This was facilitated by the greater simulated
interception storage capacity of the thicket forest (Fig. 4b) and is consistent with findings in relation to commercial plantations
(e.g. Birkinshaw et al., 2014; Farley et al., 2005; Johnson, 1990). Interestingly, the greatest increase in interception evaporation
occurred during the dormant season, resulting in a proportially larger increased in "green" fluxes at this time rather than
during the more biologically active period. This has been observed in other studies where the canopy is wet for prolonged
periods over winter (e.g. Birkinshaw et al., 2014; Peng et al., 2016), and seems to reinforce the importance of enhanced
turbulent airflows over forests and sensible heat exchanges between the canopy and atmosphere, in facilitating high
interception evaporation when available net radiation is reduced (c.f. Stewart, 1977; Gash and Stewart, 1977). To a lesser
degree, simulated increases in summer transpiration also contributed to greater apportionment of water to "green" fluxes (Table
3). The reduced importance of increased transpiration relative to interception evaporation has been observed elsewhere for
coniferous forests (Farley et al., 2005; Marc and Robinson, 2007).

Increased losses to interception evaporation and transpiration were slightly compensated by a reduction in simulated soil
evaporation (Table 3) that was also reflected in more isotopically depleted summer stream flows (Fig. 5c). Overall, however,
availability of water for GW recharge and streamflow was reduced, notably resulting in lower simulated summer baseflows
(Fig. 5b; c.f. Iacob et al., 2017). This likely reflected increased transpiration demand (Fig. 8b) amplifying the effect of reduced
GW subsidies from regenerating areas on the podzols and rankers to the valley bottom (Fig. 7e; Brown et al., 2005; Calder,
1993). In the BB, these subsidies are crucial for recharging GW stores in drift deposits that sustain baseflow conditions,
particularly in winter when GW recharge is usually highest (Table 3; Blumstock et al., 2016; Kuppel et al., 2020). This finding
is significant as it reinforces the need to consider the wider hydrological consequences of regeneration occurring in specific
areas such that the "right tree [is planted] in the right place" (Forestry Commission Scotland, 2010). Greater consistency in
simulated high flows suggests that increases in storage capacities (Fig. 4) and "green" water fluxes (Table 3) for the thicket
scenario were insufficient to moderate the combined influences of antecedent conditions and precipitation inputs that led to
the largest events modelled here (Fig. 5b). This is consistent with previous work showing forest regeneration may have a
limited impact on the magnitudes of the largest flood events (Calder et al., 2007; Iacob et al., 2017; Soulsby et al., 2017).





Greater alignment between simulated water partitioning dynamics in the baseline and old open forest scenarios (Table 3; Figs. 7 and 8) was consistent with previous work in plantations that has suggested the hydrological impacts of forests will lessen as

they mature (e.g. Delzon and Loustau, 2005; Du et al., 2016; Marc and Robinson, 2007). However, the simulated drying out of the valley bottom in the thicket scenario (Fig. 7) raises the question as to whether pinewoods could progressively encroach downslope to replace the bog woodland on the peaty gleys and, eventually, the *Sphagnum* on the peat. This would entail the creation of a positive feedback loop whereby establishment of forest reduces downslope waterlogging to permit further forest expansion and, ultimately, the shift of the catchment into a different state of dynamic equilibrium (c.f. Rodriguez-Iturbe, 2000;

Peterson and Western, 2014; Peterson et al., 2009). In turn, this could prolong the changes to water partitioning simulated for the thicket scenario.

### 5.2 Storage-flux dynamics of regenerating forests as revealed by water ages

Given uncertainty in simulated water ages, it is difficult to definitively conclude how forest regeneration would affect source

waters of "blue" and "green" fluxes. In general, increased losses of zero-aged precipitation to interception evaporation in the thicket scenario created a tendency for slower turnover of below-canopy water that was particularly expressed in the older average ages of soil evaporation, lateral GW flows and streamwater (Table 4). The greater increase in soil evaporation age relative to L1 age likely reflected reduced evaporation of younger water from the freely draining soils on the hillslopes increasing the dominant influence of evaporation of well-mixed, older water from the valley bottom (Figs. 6 and 8c; Sprenger

et al., 2017; Tetzlaff et al., 2014). It may also indicate that periods of greatest atmospheric evaporative demand were satisfied by soil waters from the valley bottom on which the aging effects of older upslope GW subsidies were most imprinted. Average streamflow ages reflected the potential for low/moderate flows to consist of older water (Fig. 5d) which, in turn, indicated increased relative contributions of GW that was itself older. Streamwater ages remaining relatively young during higher flows supports the previous assertion that regeneration did not prevent activation of rapid OLF paths in larger events. Average ages

of these fluxes were generally restored in the old open forest scenario (Table 4).

The similarity in average transpiration ages amongst simulated scenarios contrasts with other studies in drier catchments that have suggested forests and other vegetation covers, such as grassland, transpire water with differing age characteristics (Douinot et al., 2019; Smith et al., 2020). This suggests that the wet, low energy climate of the BB and generally well-mixed

nature of hydrological stores allows the catchment to accommodate increased "use" of water by forests more readily than drier environments with less retentive soils in which forest water uptake must be satisfied by younger, more recent inputs of water that increases their susceptibility to drought-induced water stress (Kleine et al., 2020; Smith et al., 2020). Reduced 95th percentile average ages in the thicket scenario (especially for Oct-Mar) may have partially reflected greater young water contributions to forest transpiration owing to increased percolation of younger soil water from L1 to L2 as a consequence of

enhanced root water uptake from the latter (c.f. Table S2; Fig. 4a). However, that transpiration ages remained on the order of





months to years implies that moisture carried over from previous seasons was sufficient to satisfy transpiration demand (c.f. Allen et al., 2019; Brinkmann et al., 2018; Kuppel et al., 2020). Slightly older median transpiration ages in the old open forest scenario likely reflected increased uptake from the lower soil layers (Fig. 4a) combined with transpiration fluxes that had a more limited effect on water turnover rates relative to the thicket scenario (Table 3i).


**5.3 Implications of regeneration for hydrological source areas and connectivity**

The effect of forest regeneration on hydrological source areas and connectivity can be informative regarding its use as a nature-based solution to management of water quantity and quality. Consistent with inferences from the discharge timeseries (Fig. 5b), establishment of thicket forest most strongly affected spatial patterns of connectivity for the examined low and moderate

summer flow events (Fig. 9a-d), whilst only relatively minor changes were simulated for a large winter event (Fig. 9e-f). Interestingly, whilst daily average OLF was simulated to reduce across much of the catchment (Fig. 7a), examination of event-based connectivity revealed that only specific flow paths or areas of the catchment (e.g. SW headwaters) were likely to fully disconnect from the stream. This highlights that certain areas may be more sensitive to forest establishment and could be useful for managing regeneration to minimise/maximise its impact on certain flow types (Collentine and Futter, 2018; Iacob et al.,

2017). The inability of regeneration to fully interrupt connectivity between the hillslopes and riparian zone, a major driver of non-linear storm flow responses (Birkel et al., 2015; Soulsby et al., 2015; Stockinger et al., 2014), under increasingly wet catchment states additionally offers an explanation as to why high flow magnitudes tended to be maintained across the simulated scenarios. This may also limit the effectiveness of forest regeneration in regulating certain water quality parameters by reducing redistribution of contaminants from the hillslope to the riparian zone (e.g. faecal indicator organisms – Neill et al.,

565 2019).

Whilst these findings imply that forest regeneration may "slow the flow" (Fig. 7a) but not fully disconnect surface flow paths from the stream under increasingly wet conditions (Fig. 9), it is likely that simulated changes to connectivity are conservative. This is because increases in surface roughness and detention storage caused by vegetation change cannot presently be simulated

in EcH$_2$O-iso; however, these factors may also affect connectivity alongside changes to water flux partitioning (Collentine and Futter, 2018; Turnbull and Wainwright, 2019).

**6 Conclusions and wider implications**

In this work, we demonstrate that the ecohydrological consequences of natural forest regeneration on degraded land depend on the structural characteristics of the forest at different stages of development. We also show how hydrological functioning

of the wider catchment can be affected by spatial changes in water flux partitioning caused by regeneration in specific areas (e.g. hillslopes). Consequently, land cover change studies need to move beyond simply considering forested vs. non-forested



scenarios to provide a robust evidence base for management decisions seeking to balance regeneration/rewilding with other ecosystem services. Early stages of regeneration are suggested to have the most significant effects on catchment hydrology. The drier catchment state and reduced low/moderate flows may have negative implications for fire risk (Turetsky et al., 2015),

aquatic ecosystems (e.g. those supporting Atlantic Salmon populations – Moir et al., 1998) and downstream services like drinking water provision. However, recovery of flux partitioning and water ages during later stages of regeneration implies that such issues may be transient, with landscapes covered by older forest able to support pre-existing ecosystem services whilst improving biodiversity. Potential effectiveness of regeneration as a nature-based solution to water quantity and quality issues appeared greatest during the thicket stage for dry to moderately wet catchment states. Whilst the impact on high flow

magnitudes was limited for events simulated here, it is possible that the frequency of such events could be reduced; however, assessment would require a longer run of data to derive flow duration curves (Alila et al., 2009).

Our work also highlights the value of tracer-aided ecohydrological models as tools for land cover change investigations. In particular, processes such as enhanced forest interception evaporation could be explicitly simulated (c.f. Calder, 1976), whilst

successful reproduction of diverse ecohydrological and isotope observations increased confidence in simulated catchment internal functioning. Such models also have further potential. In the first instance, incorporating temporal variability in soil characteristics could provide a more complete understanding of the ecohydrological consequences of land cover change; however, there remains a need to better characterise how soil properties change in response to land cover (Archer et al., 2013; Chandler et al., 2018) and to understand how this translates into changes in "effective" model parameters (Seibert and van

Meerveld, 2016). Second, whilst static snapshots were used to simulate stages of land cover change, ecohydrological models could potentially explore the development of dynamic feedbacks that could alter trajectories of change (e.g. forest encroachment into the valley bottom; c.f. Perino et al., 2019; Scott and Prinsloo, 2008). For scenarios of vegetation change, this would be contingent on processes such as seed dispersal and species competition being conceptualised in models such as EcH$_2$O-iso (c.f. Fatichi et al., 2016).

**Code and data availability**

Data are available from the corresponding author on request. The model code for EcH$_2$O-iso is available at: https://bitbucket.org/sylka/ech2o_iso/src/master_2.0/

**Author contribution**

AJN and CS formulated the overarching research goals and approach of the paper. AJN, in discussion with CS, carried out
model setup, calibration, scenario analysis and visualisation of results, using the EcH$_2$O-iso model developed as part of work by DT, CS and MPM. All authors contributed to interpretation of results. AJN drafted the initial manuscript with all co-authors



extensively contributing to its evolution. CS, along with CB and MPM, secured funding for the ISOLAND project, and was responsible for oversight of research activities. DTs VeWa project facilitated original data collection for the Bruntland Burn.

**Competing interests**

The authors declare that they have no conflict of interest.

**Acknowledgements**

Funding for this work was provided by the Leverhulme Trust ISOLAND project (RPG 2018 375) and the European Research Council VeWa project (VeWa project GA 335910). Many thanks go to past members of the VeWa project who were involved in collection and curation of data for the Bruntland Burn catchment. All model runs were carried out on the Maxwell High
Performance Cluster (HPC) funded and maintained by the University of Aberdeen.

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
