# Peer review of "Structural changes to forests during regeneration affect water flux partitioning, water ages and hydrological connectivity: Insights from tracer-aided ecohydrological modelling"

_Hydrology and Earth System Sciences, 2021_

## Referee Comment (RC1)

**hess-2021-158 review**

The authors present a modelling study analysing the effect of forest regeneration on blue and green water fluxes for a catchment in the Scottish Highlands, which have undergone dramatic decreases of native pinewoods since the 17th century. The authors use the tracer-aided ecohydrological model EcH2O-iso (Kuppel et al., 2018a) to model flux partitioning, water ages and hydrological connectivity under three different conditions (i.e., baseline conditions, thicket forest and old-open forest) representing different stages of natural forest regeneration.

The model results highlight that the thicket forest stage leads to the greatest changes in flux partitioning, water ages and hydrological connectivity especially during low flow, while establishment of old-open forest will likely result in the system returning to similar ecohydrological fluxes as during baseline conditions.

The authors argue that this study demonstrates the importance of considering different stages of regeneration as well as their spatial and temporal impact on ecohydrological partitioning to accurately inform landscape restoration.

**General comments**

The study fits the scope of Hydrology and Earth System Sciences and represents an important contribution to investigating the effect of landscape restoration. The study uses existing concepts and methods, but applies them to different landscape scenarios than previous research. Hence, the paper represents a substantial contribution to scientific progress in this field.

The paper is well-written and considers an appropriate amount of related work. The figures and tables are well chosen to support the results and conclusions of the study.

I do not have major general comments, but I am missing some more in-depth discussion as to (1) the added value of the isotope module and (2) the likelihood of the two land-cover change scenarios under climate change. For (1), the authors refer to the validation by Kuppel et al. (2018a), but it would be useful to discuss in the paper to what extent the isotope data helped constrain model parameters and whether the model parameters sensitive to the isotope data are crucial for this study. In view of the uncertainty bounds in the behavioural solutions and to illustrate the value of the isotope data, the authors might want to include a baseline simulation without isotope data and compare the model uncertainties to those of the tracer-aided simulation. Related to this is also the discussion of changes in water ages with progressing regeneration (section 5.2), which should underline more why this information is highly beneficial for assessing regeneration changes as opposed to looking at the changes in blue and green fluxes only (and thus why we need the isotope data).

Regarding (2), given that the full regeneration to old-open forest might take several decades, I am wondering whether changing climate might lead to a different trajectory of change than the one depicted in the study. More specifically, how realistic is it that the system can meet increased evaporative demand during summer (e.g., Werritty and Sugden, 2013)? Would it be possible to test this for the study catchment with the EcH2O-iso model (see page 30, lines 595–599)?

I also have a comment on the data availability. According to the HESS data policy, "data and other information underpinning the research findings are "findable, accessible, interoperable, and reusable" (FAIR) not only for humans but also for machines". If the data cannot be made publicly available, there

should be "a detailed explanation of why this is the case". Please provide the data in an open repository or explain why this would not be possible.

**Specific comments**

- page 6, line 150: do you mean that there is an exponential decrease of roots in each layer with depth? Please clarify.
- page 6, line 160: could you briefly comment on the impact of this assumption of complete mixing? With a total soil depth of around 30 m in some simulations, how does this assumption affect the water age simulation? I could imagine that the L3 soil layer might contain a relevant proportion of older water, which might bias the water age of transpiration towards older ages using the complete mixing assumption.
- page 6, line 168: "soil types were assumed to be spatially uniform". I am not sure I understand. Do you mean there is only one soil type per cell (as in Fig. 1a) or what exactly is spatially uniform? Also it is not clear to me how to read Table 1: should the percentages across all vegetation types (including bare soil) for each soil type add up to 100%? Could you explain this in a bit more detail in this paragraph?
- page 9, line 214: how many simulations meet the criterion of simulated saturation areas < 60%? Why are only 30 runs of those retained as behavioural results? This is probably a small proportion of the first subset, but it still gives large uncertainty bounds, for example, in flux ages.
- page 12, lines 282–283: I do not fully understand. What kind of threshold and what is the role of reinfiltration along a flow path? Please clarify.
- page 13, lines 290–291: could you also state the values of the performance metrics for behavioural runs?
- page 27, lines 507–510: "Greater consistency…". I am not sure I understand. Do you mean that regeneration does not affect the fluxes during larger events because of sufficient amount of rainfall and stored pre-event water during these events?
- page 28, lines 517–520: So would that mean that the old forest state might be achieved much later or maybe not at all?
- page 29, lines 554–555: I do not see big differences in the connectivity changes between low / moderate summer events and the large winter event. Could you support this assertion by mentioning percentage changes in section 4.7?
- section 5.3: see general comment (2): I would appreciate some words on the likelihood that regeneration would undergo these two forest stages in view of climate change. Could it be that less rainfall/higher ET in summer would lead to a diversion of pinewood regeneration as depicted in Fig. S1 such that increased transpiration demand of thicket forest could not be met and transition to old forest would not occur? This links to the statement on exploring trajectories of change made in the Conclusions.
- page 30, line 590: see general comment (1): I am not sure about the benefits of the isotope observations here. Do we need the isotope module of the model or what additional validation data might be useful to constrain the uncertainty bounds? Could the authors comment on the uncertainty that would result from calibration without isotope data? Is it the comparably low temporal and spatial resolution of soil-water isotope data that limits the uncertainty reduction?

**Technical corrections**

- page 5, line 142: gridded

**Tables**

- Table 1: how did the authors determine the exact proportional aerial coverage in the two scenarios? References are given here but it is not clear to me whether/how these numbers have been derived from the information provided in the references

**Figures**

- Figure 1a: would it make sense to have more meaningful symbols and/or colours for the monitoring sites, grouping weather stations, soil types and vegetation?
- Figure 1: could you also include a digital elevation model so it is easier to see the location of the hillslopes in the catchment?
- Figure 9: could you also show the dates of the different snap shots directly in the figure? If not, the reader has to switch back and forth between the figure panels and the figure caption.

…………………………………………………………

**References used in this review**

Kuppel, S., Tetzlaff, D., Maneta, M.P., Soulsby, C. (2018a). EcH2O-iso 1.0: water isotopes and age tracking in a process-based, distributed ecohydrological model. Geoscientific Model Development 11: 3045-3069. DOI: 10.5194/gmd-11-3045-2018

Werritty, A., & Sugden, D. (2012). Climate change and Scotland: Recent trends and impacts. *Earth and Environmental Science Transactions of the Royal Society of Edinburgh, 103*(2), 133-147. doi:10.1017/S1755691013000030

---

## Referee Comment (RC2)

**General comments and recommendation**

The manuscript by Neill et al. presents an ecohydrological modelling study about structural changes of forest regeneration and the effect on water flux portioning, water ages and hydrological connectivity. They use the EcH$_2$O-iso for a small experimental catchment in the Scottish Highlands, and simulate a baseline and two land cover change scenarios, a thicket and an old-open forest.

The modelling study gives the opportunity to create an old-open forest which might be very difficult to create in a field experiment due to long time period over 100 years, and agriculture forest use (tree age around 40 years). This stage of old forest might happen if the forest harvesting stops, hence especially for stakeholders it is interesting to see the influence of such forest development. But also, the research community gets an idea about the effects of a thicket and an old-open forest to the hydrological conditions. This could help to see the field experiments with a different angle and to support information around such experimental sides.

The text is well structured with meaningful subheadings and well-structured paragraphs. The manuscript is in the scope of the HESS journal and gives new insights in the field of tracer-aided ecohydrological modeling.

I see an especially need to strengthen the text for an easier readability with less abbreviations and clear sentences. The figures and table also need some revisions for an easier readability, e.g. bigger fonts. Here I give some general comments and specific comments at the end. ("Line" is abbreviated with "L")

**General comments**

- Abbreviations
  - I suggest to reduce the abbreviations for easier and an undisturbed readability. Especially since some abbreviations are just used a few times (e.g. SW 4x, VWC – 6x, RZ – 7x, OLF-13x). From my point of view, I would only keep LAI and use the full words for the others. ET and GW, might be an option to keep as well, but it still interrupts the reading.
  - (As an alternative, a table with all the abbreviations could also work)
  - Some abbreviations are not introduced in the text e.g. NE (Line 286), SE (Line 463), NW (Tab1, L 385, L403)
  - Leaf area index is mentioned in L 93, but introduced in L 186, I would also suggest not to introduce the LAI in italic, or if this is really necessary only use the italic version, also for figures and tables
  - Bruntland Burn, I would suggest to keep the whole name instead of just BB

- Figures
  - In general, the figure captions are quite short, maybe some more information for the reader to understand the meaning of each figure can be added.
  - Fig. 1a:
    - the symbols should be the same for the same type of station e.g. gauging station in the river (one symbol), weather station (another symbol), ect.

- DW abbreviations should be explained (unclear for me what it could be)
- Map in the left corner is too small, better to use a bigger map with some parts of Europe to show persons from everywhere, where the catchment is located
- The whole figure looks a bit unstructured and a bit chaotic, maybe it is better to split in two figures
- 1g) Bog pine, it seems that there is no bog pine at all, is that right? Order the scaling of the vegetation fraction to undifferentiated
- The font size of "Vegetation fraction" including number is too small, also the legend of h),
- The font size of the headings of c) to g) could be little bit bigger
- In Tab 2, Groundwater wells are mentioned. Maybe you can also include the location in this figure

- Fig. 2:
  - What is "Regen-baseline", better (regeneration – baseline scenario), since there is enough space to write the full text
  - Maybe it is an option to include Fig 1 c) to g) in Fig 2, to reduce the overloaded Fig 1.

- Fig. 3:
  - Please add a legend to every subfigure, starting with first observation, second spread, or the other way around.
  - b) (In $m^3 \, s^{-1}$), guess it is just ($m^3 \, s^{-1}$)
  - font size could be a bit bigger, for easier readability
  - For the caption I would suggest: a) Precipitation; b)  observed and simulated Discharge; c) […]

- Fig. 5:
  - Discharge again (**In** $m^3 \, s^{-1}$)
  - What is the brown color? The red on top of green? This is hard to see, even for a non-color-blind person (maybe you can find other colors e.g. red and green is not visible for many persons)
  - Caption maybe: c) Stream **water** …

- Fig. 6:
  - "Baseline:" It is better mentioned it in the Figure caption, but not as a heading, if it is always the same for all cases.
  - For comparison, it would be much easier to read and compare the subfigures, if the "spread"- median daily average would always be the same size. e.g. from 0 to 30 or so for the blue ones and 0 to 2.5 for the green ones.
  - I would also suggest to write groundwater instead of GW and evapotranspiration instead of ET, since it is enough space to write the full word.
  - font size should be a bit bigger, for easier readability
  - what are the brown pixels in a) and e), please explain e.g. in the figure caption
  - Caption: please define the abbreviation "L1"

- Fig. 7:
  - What are the brown pixels in a) and e), please explain e.g. in the figure caption
  - To get an easier overview I would suggest to write the month in the middle over the first and second subfigure column, and the third and fourth subfigure

column, since they are always showing the same time frame, just the scenarios are different.

- Again, please use the same spread for all figures maybe 0.5 to -9
- Caption: e) GW flow instead of Groundwater flow

- Fig. 8:
    - Again, please use the same spread for all figures maybe +1 to -1
    - Again, I would suggest to write the month in the middle over the first and second subfigure column, and the third and fourth subfigure column.
    - font size should be a bit bigger, for easier readability
    - for an easier overview you might consider to include the timing, so 22 July 2013, 10 August 2014 and 30 December 2015 or dry summer period, summer wet period and 100-year return period flood, or something like this
- Fig 9:
    - The figure caption does not fit on the same page, so the figure must be small, but it is no option to just minimize the total figure, since already now the text and numbers are very hard to read

- Tables
    - In general, the table captions are quite short, maybe some more information for the reader to understand the meaning of each figure can be added.
    - Tab 1:
        - The whole "cover" column should be left-justified, or why is only "A baseline" right-justified?
        - Instead of just "cover" maybe "proportional aerial coverage" or something like this
        - The use of italic is confusing, maybe use bold instead of italic for "Baseline", "Thicket woodland", …
        - Notes a): "pre-existing" with small letter
    - Tab 2:
        - What does "Full" stand for? Full time period? Then maybe also mention again how long this study period is or from x to x.
        - I would suggest to only use "and" or "&", not both in the same table
        - You might want to explain the A and B behind Forest and Heather
        - Where is the location of "deeper well" 1 to 4, maybe include in Fig 1.
    - Tab 3:
        - Is it really necessary to give the decimal place, full numbers are easier readable (like done in Tab 4)
        - What is the added valued to include the second columns with the differences in seasonally averaged flux totals → The table is quite confusing, so maybe it is better shorten the given information, if possible (this also applies for Tab 4)

- Words
    - please stick to one version of "old-open" or "old open" including the abstract, tables and figures
    - instead of Oct-Mar and May-Sep I would introduce the words of summer and winter or, dormant season and biological active season, beside a better readability this might also

be an improvement of the figures, if you want to stick to the month, I would suggest to write the full names like October to March.

- Introduction
  - L 95 – 99 You explain, that the soil properties are held constant, but then further describe that they might change. I guess, it would be very interesting to see the effect of soil property changes. How strong is the effect here?
  - Also, climate change has an important impact to the soil and plants, especially in a 100-year scenario. Maybe you can further explore this part in the introduction or later on.

- 3.1 The EcH2O-iso model
  - The model description part with its concept is a bit imprecise.
  - The kinematic wave model in the groundwater context (L.154 -160) is not so common, it is normally known for open channel routing. Maybe you can explain this point a bit more detailed. From the description, the term GW is maybe not the right one in L 157, maybe it is interflow? Is there an exchange of river and groundwater (in one or both directions)?

- Calibration
  - Give less references to the specific parts of the figures and table. e.g. L339 to 342 (Tab 3) at the end of the sentence is enough. There are many other places where the references to Tables and Figures can be reduced for a much easier readability, without losing information. (e.g. L345, L 346, L 355 (when the whole paragraph is about the figure introduces at the beginning it is not necessary to refer to all the subfigures after each sentence.)
  - 4.1 Baseline calibration: refer more to the Table 2, e.g. with the MAE for discharge.

- Discussion
  - Sometimes difficult to read, especially the very long sentences: L 480 – 483, L 491 – 494, L 507 – 510, L 542 – 545, L 560 – 563
  - Here you introduce the terms of dormant season and biological active season (L490 – 491), and winter and summer (e.g. L 505), but without giving the month you refer to in you catchment.

**Specific comments**

L 39 – 42: very long sentence, please split in two

L 82: maybe delete "which"

L 90: maybe give the catchment area in brackets, and not only call it small

L94 – 99: changes in soil properties are not included in the model, but here explained that it is very likely to happen. Why are you not including soil property changes when you think they are happening and important? I guess it needs more thoughts why you did not include them. Also, a connection from the missing soil property changes to the specific objectives of the manuscript would be helpful.

L 106: reference to Fig 1, not only Fig 1a, the whole figure gives information about the catchment

L 116: (SNH, 2016) instead of [SNH, 2016]

L 123: Maybe better: Mean annual precipitation is 1000 mm and potential evapotranspiration is 400 mm, with the […]

L 125: Maybe better: […] mean temperatures ranging between 1 °C in winter and 13 °C in summer.

L 128: please include catchment after BB, also in the other cases in the manuscript so "… BB catchment" e.g. L 197, L 210, …

L 149: please explain the soil layer L1, L2 and L3. Is the L1 the top most? How are they defined, maybe with the soil horizons? Or just with a given depth?

L 155: please give the source of the Green-Ampt model

L 168: what is meant by "spatially uniform", please describe further

L 182: better: 100 m x 100 m grid

L 183: add "in the supplementary Table S1." Or something similar, to know where to find the table, since it is not in the manuscript itself. Also, at other places when referring to the supplementary material e.g.  L 185, L 190, L221, …

L 188: What kind of channel? River channel?

L 207: "to avoid over-emphasising high flows" – compared to what? Compared to NSE?

L 271: add … *periods of biological growth and dormancy in our study area.* Or something similar

L 289: model skills instead of model skill

L 291: Tables 2 and S2, since the supplementary, should just give additional more detail information, so is less important and should be mentioned as a second.

L 369: "zero" instead of "0"

---

## Author Response (AR1)

Dear Dr Hildebrandt,

Thank you for giving us the opportunity to revise our manuscript. We have addressed both editor and reviewer comments as detailed below and think that the manuscript is now much stronger as a result. In response to comments from both reviewers, we have especially improved the discussion of uncertainty in simulated regeneration scenarios (e.g. due to climate change) and the model more generally. In addition, we have added greater nuance to our findings related to the effect of regeneration on streamflow and connectivity, and further strengthened our interpretations of water ages to better highlight why their consideration is particularly useful. The presentation of Tables and Figures has also been refined throughout and we have endeavoured to improve readability.

Whilst it was suggested we should consider a dedicated analysis of how isotopes benefit tracer-aided model calibration, we did not feel this was within the scope of the current paper. Consequently, we have removed any claims that our work **demonstrates** how use of such models improves confidence in simulated catchment functioning. In the Introduction, however, we did strengthen references to work where this value has been previously established. We hope this suitably addresses the concerns of Reviewer 1 (who recommended acceptance subject to minor revision) so that a further round of reviews can be avoided.

We greatly appreciate the efforts made in helping us to improve our manuscript which we hope is now acceptable for publication in HESS.

Kind regards,

Aaron Neill, on behalf of all co-authors.
* * *
**Author responses to Editor Comments**

*EC-1:* I regret that no interactive discussion ensued on the main concern (1) of reviewer 1 in order to avoid a second round of review. If you can manage to accommodate this concern in the best possible way, you may still be able to avoid this time consuming step. I noted that in the conclusion and implication part (L 588 ff) the manuscript claims that it highlights the value of tracer-aided models. I agree with reviewer 1 that this claim cannot be made simply by referring to the literature. I therefore suggest that either a dedicated analysis is included, or this conclusion is removed.

*Response to EC-1:* As we did not feel a dedicated analysis was within the scope of this paper, we opted to remove the conclusion regarding the value of isotopes in improving confidence in tracer-aided model calibration. In the Introduction, we did, however, strengthen references to previous studies (one of which was in our catchment) where incorporation of isotopes in model calibration helped to better constrain storages and mixing volumes to enable more consistent simulation of catchment hydrological

functioning. We hope that this helps to suitably address the concerns of Reviewer 1 so that a further round of reviews can be avoided.

*EC-2:* I agree with Reviewer 2 on the color scales and the font size in Figures 6-8. Especially in Figs 7 and 8, where differences are shown, it would be good to have an idea of which of the fluxes is most affected. Maybe consider relative differences?

*Response to EC-2:* We have increased font sizes in all Figures where possible. Additionally, we found a means of using consistent colour scales in Figures 6-8 to improve comparability of fluxes and differences whilst also preserving simulated spatial patterns.
* * *
**Author responses to RC-1**

Summary

The authors present a modelling study analysing the effect of forest regeneration on blue and green water fluxes for a catchment in the Scottish Highlands, which have undergone dramatic decreases of native pinewoods since the 17th century. The authors use the tracer-aided ecohydrological model EcH2O-iso (Kuppel et al., 2018a) to model flux partitioning, water ages and hydrological connectivity under three different conditions (i.e., baseline conditions, thicket forest and old-open forest) representing different stages of natural forest regeneration.

The model results highlight that the thicket forest stage leads to the greatest changes in flux partitioning, water ages and hydrological connectivity especially during low flow, while establishment of old-open forest will likely result in the system returning to similar ecohydrological fluxes as during baseline conditions.

The authors argue that this study demonstrates the importance of considering different stages of regeneration as well as their spatial and temporal impact on ecohydrological partitioning to accurately inform landscape restoration.

*Response to Summary:* Thank you to the reviewer for taking the time to read our work and provide constructive comments to strengthen the manuscript.

General comments

*RC-1.1:* The study fits the scope of Hydrology and Earth System Sciences and represents an important contribution to investigating the effect of landscape restoration. The study uses existing concepts and methods but applies them to different landscape scenarios than previous research. Hence, the paper represents a substantial contribution to scientific progress in this field. The paper is well-written and considers an appropriate amount of related work. The figures and tables are well chosen to support the results and conclusions of the study.

*Response to RC-1.1:* Thank you for recognising the significance and quality of our work.

I do not have major general comments, but I am missing some more in-depth discussion as to (1) the added value of the isotope module and (2) the likelihood of the two land-cover change scenarios under climate change.

*RC-1.2:* For (1), the authors refer to the validation by Kuppel et al. (2018a), but it would be useful to discuss in the paper to what extent the isotope data helped constrain model parameters and whether the model parameters sensitive to the isotope data are crucial for this study. In view of the uncertainty bounds in the behavioural solutions and to illustrate the value of the isotope data, the authors might want to include a baseline simulation without isotope data and compare the model uncertainties to those of the tracer-aided simulation.

*Response to RC-1.2 & RC-1.16:* We respond jointly to this comment and RC-1.16 given that they are related. Thank you for the suggestion of running a baseline simulation without isotope data; however, we did not feel this was within the scope of this paper. Consequently, and in response to EC-1, we have removed the conclusion that our paper showed the value of using isotopes in calibration of tracer-aided models to improve confidence in simulated catchment hydrological functioning. We did, however, strengthen references to past studies (one of which took place in our catchment – Birkel et al., 2011) where this value has previously been established to better highlight why we used a model such as EcH$_2$O-iso in this work (L87-91).

We also now more explicitly note that some model outputs demonstrated larger uncertainties despite the rich calibration dataset used in our study and suggest reasons as to why this might be (L513-521). Overall, however, we argue that the model still provides a useful tool for simulating the effects of forest regeneration at present (L522-525). The need for further research into how models such as EcH$_2$O-iso can be better constrained is also now highlighted in Sect. 6 (L673-674).

*RC-1.3:* Related to this is also the discussion of changes in water ages with progressing regeneration (section 5.2), which should underline more why this information is highly beneficial for assessing regeneration changes as opposed to looking at the changes in blue and green fluxes only (and thus why we need the isotope data).

*Response to RC-1.3:* Interpretations of water ages (Sect. 5.2) were strengthened to better demonstrate how they can enhance understanding of spatial (e.g. L580-587) and temporal (e.g. L588-589) aspects of catchment storage-flux interactions and their sensitivity to change (e.g. L590-593). In addition, implications of water age dynamics for wider ecosystem resilience were highlighted (L597-601).

*RC-1.4:* Regarding (2), given that the full regeneration to old-open forest might take several decades, I am wondering whether changing climate might lead to a different trajectory of change than the one depicted in the study. More specifically, how realistic is it that the system can meet increased

evaporative demand during summer (e.g., Werritty and Sugden, 2013)? Would it be possible to test this for the study catchment with the EcH2O-iso model (see page 30, lines 595–599)?

*Response to RC-1.4 & RC-1.15:* We respond jointly to this comment and RC-1.15 given that they are related. Thank you for this comment; we agree that the discussion surrounding the likelihood of the simulated scenarios could be improved. Consequently, a new Discussion section (Sect. 5.4) was added that addresses uncertainties in the regeneration scenarios due to climate change, management decisions and the simulated drying out of the valley bottom in the Thicket scenario (previously this was in the last paragraph of Sect. 5.1). Testing the possible effects of climate change on regeneration trajectories was not possible in this paper as vegetation dynamics were switched off to minimise the number of processes requiring constraint when only simulating "snapshots" of regeneration. However, improvements that could be made to models such as EcH$_2$O-iso for exploring different trajectories of change were highlighted in Sect. 6 (L674-677).

*RC-1.5:* I also have a comment on the data availability. According to the HESS data policy, "data and other information underpinning the research findings are "findable, accessible, interoperable, and reusable" (FAIR) not only for humans but also for machines". If the data cannot be made publicly available, there should be "a detailed explanation of why this is the case". Please provide the data in an open repository or explain why this would not be possible.

*Response to RC-1.5:* Files and scripts necessary for reproducing and analysing model outputs used in this work are now available via the University of Aberdeen PURE repository (https://doi.org/10.20392/045cd572-ecc8-4dfd-b003-c0d0c621510e). The Code and Data Availability section has been updated to reflect this.

Specific comments

*RC-1.6:* Page 6, line 150: do you mean that there is an exponential decrease of roots in each layer with depth? Please clarify.

*Response to RC-1.6:* L155-158 were changed to "Transpiration demand is satisfied by root water uptake from three soil layers (L1-3, with L1 being the topmost layer) with calibrated depths, in proportion to the water content and fraction of roots in each layer. The latter is determined by an exponential function describing how root fraction decreases with depth (Kuppel et al., 2018b)."

*RC-1.7:* Page 6, line 160: could you briefly comment on the impact of this assumption of complete mixing? With a total soil depth of around 30 m in some simulations, how does this assumption affect the water age simulation? I could imagine that the L3 soil layer might contain a relevant proportion of older water, which might bias the water age of transpiration towards older ages using the complete mixing assumption.

*Response to RC-1.7:* Given the wet, low energy environment of the BB, the complete mixing assumption is likely a reasonable approximation under most conditions given the model grid size and

daily time steps. Previous empirical work in the catchment has shown limited evidence of ecohydrological separation beyond minor evaporative enrichment of increasingly mobile waters in the upper soil (e.g. Geris et al., 2015); consequently, the main limitation of the complete mixing assumption is likely to be over-enrichment of water in L1 which may be translated throughout the soil profile (Kuppel et al., 2018) rather than adverse effects on transpiration ages. The effect of L3 on the latter will further be mediated to some extent by the more limited presence of roots in this layer (most vegetation types have roots within 20-50 cm of the surface, most of which will often fall in L1 and L2).

*RC-1.8:* Page 6, line 168: "soil types were assumed to be spatially uniform". I am not sure I understand. Do you mean there is only one soil type per cell (as in Fig. 1a) or what exactly is spatially uniform? Also it is not clear to me how to read Table 1: should the percentages across all vegetation types (including bare soil) for each soil type add up to 100%? Could you explain this in a bit more detail in this paragraph?

*Response to RC-1.8:* L179 now reads "The properties of each individual soil type were assumed to be spatially uniform." Table 1 has also been reorganised by soil type and made to be more succinct to improve clarity. The percentages given in the table will not sum to 100% for a given soil type since they often refer to a percentage of land not covered by other vegetation types (e.g. trees). However, the total coverage of all vegetation types plus bare earth will sum to 100% - this is clarified in the caption.

*RC-1.9:* Page 9, line 214: how many simulations meet the criterion of simulated saturation areas < 60%? Why are only 30 runs of those retained as behavioural results? This is probably a small proportion of the first subset, but it still gives large uncertainty bounds, for example, in flux ages.

*Response to RC-1.9:* Approximately 11,000 simulations met the criteria of saturation area <60%. However, many of these simulations will have performed poorly with respect to the other calibration targets used as part of the multi-criteria approach. We retained 30 behavioural runs to strike a balance between the need to illustrate uncertainty in model outputs and the increased computational demand of running the model when producing spatial outputs required for the change analysis but not calibration. This is now explained (L231-232). Also see Response to RC1.2 & 1.16 for comments on the width of uncertainty bounds.

*RC-1.10:* Page 12, lines 282–283: I do not fully understand. What kind of threshold and what is the role of reinfiltration along a flow path? Please clarify.

*Response to RC-1.10:* To clarify, L305-309 were changed to: "A cell was considered hydrologically connected if overland flow (OLF) was simulated for the cell and all downslope cells along the flow path to the stream. No minimum threshold of OLF was imposed for a cell to be considered connected (c.f. Turnbull and Wainwright, 2019) as EcH$_2$O-iso simulates re-infiltration along a given flow path which can prevent upslope cells producing OLF from connecting to the stream (Maneta and Silverman, 2013)."

*RC-1.11:* Page 13, lines 290–291: could you also state the values of the performance metrics for behavioural runs?

*Response to RC-1.11:* The values for performance metrics are currently given in Table 2. Given the number of calibration targets, we were not keen on moving these values into the text to ensure that readability was maintained.

*RC-1.12:* Page 27, lines 507–510: "Greater consistency…". I am not sure I understand. Do you mean that regeneration does not affect the fluxes during larger events because of sufficient amount of rainfall and stored pre-event water during these events?

*Response to RC-1.12:* To clarify, L558-561 were changed to "The lesser impact of thicket forest on the simulated magnitude of high flows suggests that increases in storage capacities (Fig. 4) and "green" water fluxes (Table 3) were insufficient to overcome the combined influences of antecedent conditions and precipitation inputs that led to the largest events modelled here (Fig. 5b)."

*RC-1.13:* Page 28, lines 517–520: So, would that mean that the old forest state might be achieved much later or maybe not at all?

*Response to RC-1.13:* In this case, old forest would still be present on the hillslopes. However, drying out of the valley bottom could result in the presence of a younger regenerating forest rather than persistence of bog woodland/vegetation. This is further elaborated on in the final paragraph of the new Sect. 5.4.

*RC-1.14:* Page 29, lines 554–555: I do not see big differences in the connectivity changes between low / moderate summer events and the large winter event. Could you support this assertion by mentioning percentage changes in section 4.7?

*Response to RC-1.14:* Thank you for this comment. On reflection, our choice of events presented for the connectivity analysis did not best support our assertions regarding the effect of regeneration on connectivity as the August 2014 event was actually quite large with relatively wet antecedent conditions. In re-evaluating this part of the analysis, we have made the following changes:

- Sect. 3.5: Dates used in connectivity analysis were updated so that they were more representative of different flow types (L310-313).

- Sect. 4.3: It was highlighted that, proportionally (as revealed by plotting Ln(Q) in Fig. 5b), low to moderate flows in summer and during autumn/winter rewetting were most reduced in the thicket scenario. Meanwhile large winter events could somewhat "reset" the catchment towards baseline conditions, resulting in more limited divergence of subsequent winter/spring flows (L352-355).

- Figure 9 was replotted to show results of the new connectivity analysis. It was also condensed by removing the histograms and instead just plotting maps of the spatial distribution of cells connected in at least 50% of behavioural simulations, coloured by their flow path lengths.

- Sect. 4.7: Updated to accommodate new connectivity analysis and Fig. 9.

- Sect. 5.1: Text was refined to indicate that storage deficits resulted from increased summer transpiration accentuating the effect of lower GW recharge and downslope subsidies, causing a delay in rewetting and reductions in both summer baseflows and low/moderate events during summer and autumn/winter rewetting (L545-555).

- Sect. 5.3: Updated to reflect new connectivity analysis and refined for greater clarity.

*RC-1.15:* Section 5.3: see general comment (2): I would appreciate some words on the likelihood that regeneration would undergo these two forest stages in view of climate change. Could it be that less rainfall/higher ET in summer would lead to a diversion of pinewood regeneration as depicted in Fig. S1 such that increased transpiration demand of thicket forest could not be met and transition to old forest would not occur? This links to the statement on exploring trajectories of change made in the Conclusions.

*Response to RC-1.15:* Please see response above to RC-1.4.

*RC-1.16:* Page 30, line 590: see general comment (1): I am not sure about the benefits of the isotope observations here. Do we need the isotope module of the model or what additional validation data might be useful to constrain the uncertainty bounds? Could the authors comment on the uncertainty that would result from calibration without isotope data? Is it the comparably low temporal and spatial resolution of soil-water isotope data that limits the uncertainty reduction?

*Response to RC-1.16:* Please see response above to RC-1.2.

Technical corrections

*RC-1.17:* Page 5, line 142: gridded

*Response to RC-1.17:* This was corrected.

Tables

*RC-1.18:* Table 1: how did the authors determine the exact proportional aerial coverage in the two scenarios? References are given here but it is not clear to me whether/how these numbers have been derived from the information provided in the references

*Response to RC-1.18:* Notes of Table 1 now includes details of how proportional aerial coverages were derived from given references.

Figures

*RC-1.19* Figure 1a: would it make sense to have more meaningful symbols and/or colours for the monitoring sites, grouping weather stations, soil types and vegetation?

*Response to RC-1.19:* Symbols/colours of Fig. 1a were updated to better group monitoring sites.

*RC-1.20:* Figure 1: could you also include a digital elevation model, so it is easier to see the location of the hillslopes in the catchment?

*Response to RC-1.20:* Contour lines were added to Fig. 1a to improve visibility of the hillslopes.

*RC-1.21:* Figure 9: could you also show the dates of the different snap shots directly in the figure? If not, the reader has to switch back and forth between the figure panels and the figure caption.

*Response to RC-1.21:* Both flow conditions and representative date were added to the new Fig. 9.

……………………………………………………

References

Birkel, C., Soulsby, C., Tetzlaff, D., (2011). Modelling catchment-scale water storage dynamics: reconciling dynamic storage with tracer-inferred passive storage. Hydrological Processes 25: 3924-393. DOI: 10.1002/hyp.8201

Geris, J., Tetzlaff, D., McDonnell, J., Anderson, J., Paton, G., Soulsby, C., 2015. Ecohydrological separation in wet, low energy northern environments? A preliminary assessment using different soil water extraction techniques. Hydrological Processes 29: 5139-5152. DOI: 10.1002/hyp.10603

**Author responses to RC-2**

Summary

The manuscript by Neill et al. presents an ecohydrological modelling study about structural changes of forest regeneration and the effect on water flux portioning, water ages and hydrological connectivity. They use the EcH$_2$O-iso for a small experimental catchment in the Scottish Highlands and simulate a baseline and two land cover change scenarios, a thicket and an old-open forest.

The modelling study gives the opportunity to create an old-open forest which might be very difficult to create in a field experiment due to long time period over 100 years, and agriculture forest use (tree age around 40 years). This stage of old forest might happen if the forest harvesting stops, hence especially for stakeholders it is interesting to see the influence of such forest development. But also, the research community gets an idea about the effects of a thicket and an old-open forest to the hydrological conditions. This could help to see the field experiments with a different angle and to support information around such experimental sides.

The text is well structured with meaningful subheadings and well-structured paragraphs. The manuscript is in the scope of the HESS journal and gives new insights in the field of tracer-aided ecohydrological modeling.

I see an especially need to strengthen the text for an easier readability with less abbreviations and clear sentences. The figures and table also need some revisions for an easier readability, e.g. bigger fonts. Here I give some general comments and specific comments at the end. ("Line" is abbreviated with "L").

*Response to Summary:* We appreciate the careful review and positive comments. We are also grateful for the many constructive suggestions provided by the reviewer, although note that in some cases these relate to very minor details of the manuscript and its presentation; consequently, we did not find the need to accommodate all suggestions. Please find our individual responses below.

Abbreviations

*RC-2.1:* I suggest to reduce the abbreviations for easier and an undisturbed readability. Especially since some abbreviations are just used a few times (e.g. SW 4x, VWC – 6x, RZ – 7x, OLF-13x). From my point of view, I would only keep LAI and use the full words for the others. ET and GW, might be an option to keep as well, but it still interrupts the reading (as an alternative, a table with all the abbreviations could also work).

*Response to RC-2.1:* Full words rather than abbreviations are used for those appearing fewer than 10 times to improve readability. Respectfully, however, we believe the use of abbreviations occurring more frequently is justified given the manuscript length, as is use of the common abbreviations ET and GW.

*RC-2.2:* Some abbreviations are not introduced in the text e.g. NE (Line 286), SE (Line 463), NW (Tab1, L 385, L403)

*Response to RC-2.2:* These abbreviations were not originally defined as they refer to commonly used ordinal directions. However, they are now written out in full to improve readability.

*RC-2.3:* Leaf area index is mentioned in L 93, but introduced in L 186, I would also suggest not to introduce the LAI in italic, or if this is really necessary only use the italic version, also for figures and tables

*Response to RC-2.3:* LAI is now introduced in L100. Use of italics for model parameter names is noted in L202.

*RC-2.4:* Bruntland Burn, I would suggest to keep the whole name instead of just BB

*Response to RC-2.4:* Respectfully, we have choosen to keep the abbreviation BB when referring to the catchment as this has been common practice in previous papers based at this site.

Figures

*RC-2.5:* In general, the figure captions are quite short, maybe some more information for the reader to understand the meaning of each figure can be added.

*Response to RC-2.5:* Thank you for this comment, additional information has been added to captions as appropriate when responding to the specific comments for each Figure.

Fig. 1:

*RC-2.6:* The symbols should be the same for the same type of station e.g. gauging station in the river (one symbol), weather station (another symbol), ect.

*Response to RC-2.6:* Symbols have been updated accordingly.

*RC-2.7:* DW abbreviations should be explained (unclear for me what it could be)

*Response to RC-2.7:* DW refers to deeper groundwater well. This is now defined in the figure caption.

*RC-2.8:* Map in the left corner is too small, better to use a bigger map with some parts of Europe to show persons from everywhere, where the catchment is located

*Response to RC-2.8:* We have now repositioned the Scotland map and made it bigger. Respectively, however, we did not see that a more expansive map including Europe was necessary as this would likely make it more difficult to see exactly where in Scotland the catchment is located.

*RC-2.9:* The whole figure looks a bit unstructured and a bit chaotic, maybe it is better to split in two figures

*Response to RC-2.9:* Following the reviewer's later suggestion, c-g) have been moved to Figure 2.

*RC-2.10:* 1g) Bog pine, it seems that there is no bog pine at all, is that right? Order the scaling of the vegetation fraction to undifferentiated

*Response to RC-2.10:* No, this is not correct. There are some cells with a small cover of bog pine, typically less than 10%. To improve clarity, we have replotted c-g) so that 0% cover does not show (only the brown cells of an underlying base map are visible).

*RC-2.11:* The font size of "Vegetation fraction" including number is too small, also the legend of h)

*Response to RC-2.11:* Font sizes have been increased.

*RC-2.12:* The font size of the headings of c) to g) could be little bit bigger

*Response to RC-2.12:* Font sizes were increased when these sub-plots were incorporated into Figure 2.

*RC-2.13:* In Tab 2, Groundwater wells are mentioned. Maybe you can also include the location in this figure

*Response to RC-2.13:* As noted above, groundwater wells are denoted by DW1-4. This abbreviation is now defined in the figure caption.

Fig. 2:

*RC-2.14:* What is "Regen-baseline", better (regeneration – baseline scenario), since there is enough space to write the full text

*Response to RC-2.14:* Regeneration – baseline is now written in full.

*RC-2.15:* Maybe it is an option to include Fig 1 c) to g) in Fig 2, to reduce the overloaded Fig 1.

*Response to RC-2.15:* Thanks for this suggestion, Fig. 1 c-g) have now been moved to Fig. 2.

Fig. 3:

*RC-2.16:* Please add a legend to every subfigure, starting with first observation, second spread, or the other way around.

*Response to RC-2.16:* Legends have been added to all sub-figures

*RC-2.17*: b) (In m³ s-1), guess it is just (m³ s-1)

*Response to RC-2.17:* No, this is not correct – Ln denotes that it is the natural logarithm of discharge that is plotted, which improves visibility of lower flows. This is now defined in the figure caption and the "L" in the axis label capitalised for clarity.

*RC-2.18:* Font size could be a bit bigger, for easier readability

*Response to RC-2.18:* Unfortunately, font sizes have already been optimised for this figure given the large amount of data to present and, therefore, cannot be made bigger.

*RC-2.19:* For the caption I would suggest: a) Precipitation; b) and of observed and simulated Discharge; c) […]

*Response to RC-2.19:* Respectfully, we have left the caption as it is since "observed and simulated" relate to all subsequent variables after precipitation, not just discharge.

Fig. 5:

*RC-2.20:* Discharge again (In m³ s-1)

*Response to RC-2.20:* Please see Response to RC-2.17.

*RC-2.21:* What is the brown color? The red on top of green? This is hard to see, even for a non-color-blind person (maybe you can find other colors e.g. red and green is not visible for many persons)

*Response to RC-2.21:* Thank you for this comment. The brown colour was where simulations for the thicket and old open forest overlap. The colour scheme has now been changed so that it is colour-blind friendly. A note has also been made in the figure caption to indicate the colour of overlapping simulations.

*RC-2.22:* Caption maybe: c) Stream water …

*Response to RC-2.22:* This has been changed.

Fig. 6:

*RC-2.23:* "Baseline:" It is better mentioned it in the Figure caption, but not as a heading, if it is always the same for all cases.

*Response to RC-2.23:* Baseline is now only mentioned in the caption.

*RC-2.24:* I would also suggest to write groundwater instead of GW and evapotranspiration instead of ET, since it is enough space to write the full word.

*Response to RC-2-24:* Respectfully, we did not see the need to adopt this change as it was very minor relative to the effort needed for incorporation.

*RC-2.25:* Font size should be a bit bigger, for easier readability

*Response to RC-2.25:* Font sizes have been increased.

*RC-2.26:* For comparison, it would be much easier to read and compare the subfigures, if the "spread"-median daily average would always be the same size. e.g. from 0 to 30 or so for the blue ones and 0 to 2.5 for the green ones.

*Response to RC-2.26:* A common colour scale is now used for all blue fluxes. To preserve spatial patterns the upper limit was truncated; however, this only affected a small number of cells in the OLF map and the maximum simulated flux is noted in the caption. ET used its own scale whilst the constituent fluxes (transpiration, soil evaporation and interception evaporation) were plotted with the same scale.

*RC-2.27:* What are the brown pixels in a) and e), please explain e.g. in the figure caption

*Response to RC-2.27:* EcH$_2$O-iso does not simulate cell-to-cell overland or GW flow for cells containing a stream channel or outlet; consequently, an underlying brown base map is visible in such cells. This is now explained in the figure caption.

*RC-2.28:* Caption: please define the abbreviation "L1"

*Response to RC-2.28:* This was replaced with "first soil layer (L1)".

Fig. 7:

*RC-2.29:* What are the brown pixels in a) and e), please explain e.g. in the figure caption

*Response to RC-2.29:* EcH$_2$O-iso does not simulate cell-to-cell overland or GW flow for cells containing a stream channel or outlet; consequently, an underlying brown base map is visible in such cells. This is now explained in the figure caption.

*RC-2.30:* To get an easier overview I would suggest to write the month in the middle over the first and second subfigure column, and the third and fourth subfigure column, since they are always showing the same time frame, just the scenarios are different.

*Response to RC-2.30:* Respectfully, we did not feel the effort needed for such a minor change was warranted.

*RC-2.31:* Again, please use the same spread for all figures maybe 0.5 to -9

*Response to RC-2.31:* A common colour scale is now used for all sub-figures. The lower bound was truncated to preserve spatial detail; however, only a very small number of cells had differences below this limit and the largest negative difference is indicated in the figure caption.

*RC-2.32:* Caption: e) GW flow instead of Groundwater flow

*Response to RC-2.32:* This was changed.

Fig. 8:

*RC-2.33:* Again, please use the same spread for all figures maybe +1 to -1

*Response to RC-2.33:* Total ET retains its own colour scale to preserve spatial detail; however, constituent fluxes are now all plotted on the same scale.

*RC-2.34:* Again, I would suggest to write the month in the middle over the first and second subfigure column, and the third and fourth subfigure column.

*Response to RC-2.34:* Respectfully, we did not feel the effort needed for such minor changes was warranted.

*RC-2.35:* Font size should be a bit bigger, for easier readability

*Response to RC-2.35:* Font sizes were increased.

Fig 9:

*RC-2.36:* For an easier overview you might consider to include the timing, so 22 July 2013, 10 August 2014 and 30 December 2015 or dry summer period, summer wet period and 100-year return period flood, or something like this

*Response to RC-2.36:* Thank you for this suggestion. The flow conditions considered along with the representative dates analysed are now indicated in the figure.

*RC-2.37:* The figure caption does not fit on the same page, so the figure must be small, but it is no option to just minimize the total figure, since already now the text and numbers are very hard to read

*Response to RC-2.37:* The histograms were removed from this figure and maps updated to show cells that were connected in at least 50% of behavioural runs, coloured by their flow path lengths. This allows the figure to still convey very similar information but in less space.

Tables

*RC-2.38:* In general, the table captions are quite short, maybe some more information for the reader to understand the meaning of each figure can be added.

*Response to RC-2.38:* Thank you for this comment, additional information was added where appropriate when responding to the specific comments for each Table.

Tab 1:

*RC-2.39:* The whole "cover" column should be left-justified, or why is only "As baseline" right-justified?

*Response to RC-2.39:* "As baseline" is not right justified within the Cover column but instead extends across the Cover, Height and *LAI* Scale Factor columns as these factors are unchanged in the regeneration scenarios for the relevant vegetation types.

*RC-2.40:* Instead of just "cover" maybe "proportional aerial coverage" or something like this

*Response to RC-2.40:* This has been changed.

*RC-2.41:* The use of italic is confusing, maybe use bold instead of italic for "Baseline", "Thicket woodland", …

*Response to RC-2.41:* Underlining has been used in place of italics.

*RC-2.42:* Notes a): "pre-existing" with small letter

*Response to RC-2.42:* Corrected.

Tab 2:

RC-2.43: What does "Full" stand for? Full time period? Then maybe also mention again how long this study period is or from x to x.

*Response to RC-2.43:* Full stands for full study period – this is now written out in full and the dates given in the table caption.

RC-2.44: I would suggest to only use "and" or "&", not both in the same table

*Response to RC-2.44:* The table has been updated for consistency.

*RC-2.45:* You might want to explain the A and B behind Forest and Heather

*Response to RC-2.45:* A and B refer to north- and south-facing sites, respectively; however, we argue that in the context of this paper it is sufficient that A and B simply denote different sites.

*RC-2.46:* Where is the location of "deeper well" 1 to 4, maybe include in Fig 1.

*Response to RC-2.46:* Please see responses to RC-2.3.

Tab 3:

*RC-2.47:* Is it really necessary to give the decimal place, full numbers are easier readable (like done in Tab 4)

*Response to RC-2.47:* Now only full numbers are presented.

*RC-2.48:* What is the added valued to include the second columns with the differences in seasonally averaged flux totals → The table is quite confusing, so maybe it is better shorten the given information, if possible (this also applies for Tab 4)

*Response to RC-2.48:* The added value of showing differences is that they more clearly indicate whether the simulated direction of change was consistent amongst behavioural models. This may not be obvious from considering summaries of the seasonally averaged flux totals themselves. This is now explained in the captions of both tables.

Words

*RC-2.49:* Please stick to one version of "old-open" or "old open" including the abstract, tables and figures

*Response to RC-2.49:* "Old open forest" is now used consistently.

*RC-2.50:* Instead of Oct-Mar and May-Sep I would introduce the words of summer and winter or, dormant season and biological active season, beside a better readability this might also be an improvement of the figures, if you want to stick to the month, I would suggest to write the full names like October to March.

*Response to RC-2.50:* Oct-Mar and Apr-Sep are now introduced as the "Dormant" and "Active" seasons, respectively (L297). These terms are used consistently throughout the manuscript.

Introduction

*RC-2.51:* L 95 – 99 You explain, that the soil properties are held constant, but then further describe that they might change. I guess, it would be very interesting to see the effect of soil property changes. How strong is the effect here?

*Response to RC-2.51:* Here, we sought to justify our choice to keep soil properties constant by arguing that changing them may increase uncertainty in model outputs because a) it is not known how any physical changes to soil properties would be expressed in changes to effective model parameters, and b) it is unclear if/how soil properties might change under coniferous forest because there may be processes operating that counteract one another. We have rephrased L102 slightly to clarify the latter. Consequently, whilst it is desirable to account for changes in soil properties when modelling land cover change, more research is likely needed regarding how exactly properties change under different land cover types and how these changes translate into modifications of effective model parameters. This is indicated in Sect. 6.

*RC-2.52:* Also, climate change has an important impact to the soil and plants, especially in a 100-year scenario. Maybe you can further explore this part in the introduction or later on.

*Response to RC-2.52:* Thank you for this comment. To also address comments from Reviewer 1, we have added a short section at the end of the Discussion to comment on possible uncertainty in the regeneration scenarios due to factors such as climate change.

3.1 The EcH2O-iso model

*RC-2.53:* The model description part with its concept is a bit imprecise.

*Response to RC-2.53:* We apologise but it is not clear from this comment what exactly the reviewer would like us to change. In revision we have ensured that the description of EcH$_2$O-iso is as clear as possible.

*RC-2.54:* The kinematic wave model in the groundwater context (L.154 -160) is not so common, it is normally known for open channel routing. Maybe you can explain this point a bit more detailed. From the description, the term GW is maybe not the right one in L 157, maybe it is interflow? Is there an exchange of river and groundwater (in one or both directions)?

*Response to RC-2.54:* Further details on the GW routing mechanism (L166-168) and GW seepage to the stream channel (L168-169) have been provided. In EcH$_2$O-iso, only water in excess of field capacity in layer 3 can move laterally in the sub-surface. This water is conceptualised as GW and therefore this is the appropriate term.

Calibration

*RC-2.55:* Give less references to the specific parts of the figures and table. e.g. L339 to 342 (Tab 3) at the end of the sentence is enough. There are many other places where the references to Tables and Figures can be reduced for a much easier readability, without losing information. (e.g. L345, L 346, L 355 (when the whole paragraph is about the figure introduces at the beginning it is not necessary to refer to all the subfigures after each sentence.)

*Response to RC-2.55:* In revision we have tried to reduce excessive referencing to figures and tables. This has also resulted in the removal of letters that indicated "sub-tables" in Tables 3 and 4 since these are no longer necessary.

*RC-2.56:* 4.1 Baseline calibration: refer more to the Table 2, e.g. with the MAE for discharge.

*Response to RC-2.56:* OK.

Discussion

*RC-2.57:* Sometimes difficult to read, especially the very long sentences: L 480 – 483, L 491 – 494, L 507 – 510, L 542 – 545, L 560 – 563

*Response to RC-2.57:* We endeavoured to reduce the length of overly long sentences throughout with particular attention given to the sentences highlighted by the Reviewer.

*RC-2.58:* Here you introduce the terms of dormant season and biological active season (L490 – 491), and winter and summer (e.g. L 505), but without giving the month you refer to in you catchment.

*Response to RC-2.58:* Oct-Mar and Apr-Sep are now introduced as the "Dormant" and "Active" seasons, respectively (L297). These terms are used consistently throughout the manuscript.

Specific comments

*RC-2.59:* L 39 – 42: very long sentence, please split in two

*Response to RC-2.59:* This has been changed.

*RC-2.61:* L 82: maybe delete "which"

*Response to RC-2.61:* "Which" is necessary here.

*RC-2.62:* L90: maybe give the catchment area in brackets, and not only call it small

*Response to RC-2.62:* Respectively, we did not think it necessary to quote the size of the catchment here since it is introduced shortly after in Sect. 2.

*RC-2.63:* L94 – 99: changes in soil properties are not included in the model, but here explained that it is very likely to happen. Why are you not including soil property changes when you think they are happening and important? I guess it needs more thoughts why you did not include them. Also, a connection from the missing soil property changes to the specific objectives of the manuscript would be helpful.

*Response to RC-2.63:* Please see Response to RC-2.51.

*RC-2.64:* L 106: reference to Fig 1, not only Fig 1a, the whole figure gives information about the catchment

*Response to RC-2.64:* OK.

*RC-2.65*: L 116: (SNH, 2016) instead of [SNH, 2016]

*Response to RC-2.65:* This has been changed.

*RC-2.66:* L 123: Maybe better: Mean annual precipitation is 1000 mm and potential evapotranspiration is 400 mm, with the […]

*Response to RC-2.66:* This was changed as suggested.

*RC-2.67:* L 125: Maybe better: […] mean temperatures ranging between 1 °C in winter and 13 °C in summer.

*Response to RC-2.67:* This was changed as suggested.

*RC-2.68*: L 128: please include catchment after BB, also in the other cases in the manuscript so "… BB catchment" e.g. L 197, L 210, …

*Response to RC-2.68:* This has been incorporated.

*RC-2.69:* L 149: please explain the soil layer L1, L2 and L3. Is the L1 the top most? How are they defined, maybe with the soil horizons? Or just with a given depth?

*Response to RC-2.69:* L1 is the top-most layer. The depth of each layer is typically a free parameter requiring calibration (the case in this application). This is now indicated within the description of EcH$_2$O-iso (L156).

*RC-2.70:* L 155: please give the source of the Green-Ampt model

*Response to RC-2.70:* Mein and Larson (1973) is now given as the appropriate reference for the implementation of the Green-Ampt model in EcH$_2$O-iso.

*RC-2.71:* L 168: what is meant by "spatially uniform", please describe further

*Response to RC-2.71:* By this we meant that the properties of each soil type are uniform in space. We have updated L179 to read "The properties of each individual soil type were assumed to be spatially uniform."

*RC-2.72:* L 182: better: 100 m x 100 m grid

*Response to RC-2.72:* This was changed as suggested.

*RC-2.73:* L 183: add "in the supplementary Table S1." Or something similar, to know where to find the table, since it is not in the manuscript itself. Also, at other places when referring to the supplementary material e.g. L 185, L 190, L221, …

*Response to RC-2.73:* The first references to Table S1 (L198) and Figure S1 (L243) now make clear that these are supplementary and thus not in the manuscript itself.

*RC-2.74:* L 188: What kind of channel? River channel?

*Response to RC-2.74:* Yes, for the river channel – this has been clarified.

*RC-2.75:* L 207: "to avoid over-emphasising high flows" – compared to what? Compared to NSE?

*Response to RC-2.75:* Yes, this is compared to metrics based on mean squared errors, such as NSE. This has been be clarified.

*RC-2.76:* L 271: add … periods of biological growth and dormancy in our study area. Or something similar

*Response to RC-2.76:* We now indicate these as being periods of biological growth and dormancy in north-east Scotland.

*RC-2.77:* L 289: model skills instead of model skill

*Response to RC-2.77:* Skill should be singular here, but we have rewritten "model skill" as "skill of the model" for clarity.

*RC-2.78:* L 291: Tables 2 and S2, since the supplementary, should just give additional more detail information, so is less important and should be mentioned as a second.

*Response to RC-2.78:* This was changed as suggested.

*RC-2.79:* L 369: "zero" instead of "0"

*Response to RC-2.79:* This was changed as suggested.

……………………………………………………

References

Mein, R. G., & Larson, C. L. (1973). Modeling infiltration during a steady rain. Water Resources Research 9(2): 384–394.

**Additional noteworthy changes**

- Abstract has been updated in light of changes noted in Response to RC-1.14. Importance of water ages is also now emphasised through reference to transpiration ages. The conclusion regarding the value of tracer-aided models has been removed.

- The fact that rewilding may result in different trajectories of regeneration was highlighted in the Introduction (L44-45) and a more complete definition of the trajectory needed to reach old open forest was given in L247-250.

- Fig. 4 was replotted with larger font sizes.